# Unveiling Pharmacological Mechanisms of *Bombyx mori* (Abresham), a Traditional Arabic Unani Medicine for Ischemic Heart Disease: An Integrative Molecular Simulation Study

**DOI:** 10.3390/pharmaceutics17030295

**Published:** 2025-02-24

**Authors:** Doni Dermawan, Nasser Alotaiq

**Affiliations:** 1Department of Applied Biotechnology, Faculty of Chemistry, Warsaw University of Technology, 00-661 Warsaw, Poland; doni.dermawan.stud@pw.edu.pl; 2Health Sciences Research Center (HSRC), Imam Mohammad Ibn Saud Islamic University (IMSIU), Riyadh 13317, Saudi Arabia

**Keywords:** Abresham, Arabic unani medicine, *Bombyx mori*, ischemic heart disease, molecular docking, molecular dynamics, network pharmacology, pharmacophore modeling

## Abstract

**Background**: Ischemic heart disease (IHD), a leading cause of cardiovascular morbidity and mortality, continues to challenge modern medicine. *Bombyx mori* (Abresham), a traditional ingredient in Unani medicine, has shown promise in cardiovascular health, but its molecular mechanisms remain poorly understood. **Methods:** To explore the therapeutic potential of *Bombyx mori* for IHD, an integrative molecular simulation approach was applied. Network pharmacology was employed to identify the most favorable target receptor for the disease. Molecular docking simulations evaluated the binding affinities of chemical and protein-based compounds from *Bombyx mori* to the selected receptor. Molecular dynamics (MD) simulations confirmed the stability of these interactions under physiological conditions. Pharmacophore modeling identified key structural features critical for bioactivity, while in silico toxicity assessments evaluated the safety profiles of the compounds. **Results:** Key bioactive compounds from *Bombyx mori*, including Menaquinone-7, Quercetin, and Behenic acid, showed strong interactions with the target receptor, ACE2. The MD-based MM/PBSA calculations revealed the binding free energy values of Menaquinone-7 (−35.12 kcal/mol), Quercetin (−29.38 kcal/mol), and Behenic acid (−27.76 kcal/mol), confirming their strong binding affinity. Protein-based compounds, such as Chorion class high-cysteine HCB protein 13 (−212.43 kcal/mol), Bombyxin A-5 (−209.36 kcal/mol), and FMRFamide-related peptides (−198.93 kcal/mol), also displayed promising binding affinities. In silico toxicity assessments revealed favorable safety profiles for most compounds. **Conclusions:** This study positions *Bombyx mori* as a promising source of therapeutic agents for IHD. Future work should focus on experimental validation of these computational findings through in vitro and in vivo studies.

## 1. Introduction

Cardiovascular diseases (CVDs) remain the leading cause of mortality worldwide, with ischemic heart disease (IHD) being one of the most prevalent forms [1,2]. IHD, also known as coronary artery disease, results from a reduced blood supply to the heart, often due to the narrowing of coronary arteries [3,4]. Despite advances in modern medicine, the limitations of conventional therapies, including side effects and resistance, have led researchers to explore complementary and alternative medicines [5]. Among these, Unani medicine, a traditional system of healthcare widely practiced in Arabic and South Asian countries, has garnered attention for its potential in addressing cardiovascular ailments [6,7]. One notable Unani formulation, Khamira Abresham Hakim Arshad Wala, is a well-known remedy for cardiovascular diseases. Its primary ingredient, *Bombyx mori* cocoon, also referred to as Abresham, has been traditionally used in Arabic medicine to strengthen the heart and improve circulatory function [8,9]. While previous studies have hinted at the cardioprotective effects of *Bombyx mori* [10,11], they largely focused on its general pharmacological benefits without delving into the specific bioactive compounds or their mechanisms of action. Thus, the precise therapeutic potential of *Bombyx mori* cocoon for IHD at a molecular level remained unexplored. The *Bombyx mori* cocoon is a rich source of various bioactive molecules, including chemical and protein-based substances, that may contribute to its therapeutic effects [12,13]. However, the lack of detailed information regarding the specific molecules responsible for its efficacy against IHD has created a significant research gap. Understanding the interaction between these bioactive molecules and the molecular targets associated with IHD is crucial for elucidating the pharmacological basis of *Bombyx mori* and optimizing its therapeutic potential.

This study aimed to address the knowledge gap surrounding the therapeutic mechanisms of *Bombyx mori* for IHD using an integrative molecular simulation approach. Network pharmacology identified drug–target interactions and prioritized the most relevant receptor linked to IHD. Molecular docking simulations evaluated the binding affinities of both chemical and protein-based compounds from *Bombyx mori* to the selected receptor, while molecular dynamics (MD) simulations validated the stability of these interactions under physiological conditions. Pharmacophore modeling identified key structural features essential for bioactivity, offering insights into drug design. Network pharmacology provided a systems-level view, linking *Bombyx mori* bioactive compounds with multiple IHD-related targets and pathways. Among these, the receptor with the strongest association was prioritized for docking studies. The docking results highlighted promising compounds, which were further validated through MD simulations to confirm dynamic stability and biological relevance. Pharmacophore modeling revealed critical pharmacophoric features, enabling the identification of structural elements necessary for therapeutic efficacy. This study significantly advances the understanding of *Bombyx mori* cocoon as a potential therapeutic for IHD, bridging traditional Unani medicine with modern computational tools. By integrating network pharmacology, docking, MD simulations, and pharmacophore modeling, this research highlights *Bombyx mori*’s potential as a novel source for cardiovascular therapeutics.

## 2. Materials and Methods

### 2.1. Materials

The chemical compounds of *Bombyx mori* were identified through a comprehensive literature review of related journals [14,15,16], addressing the lack of a dedicated and detailed database for bioactive compounds in Unani medicine. Protein and peptide sequences derived from *Bombyx mori* were obtained through a systematic search in UniProt, a comprehensive repository of protein sequences and functional information. The search employed the MeSH term “*Bombyx mori*” within UniProtKB to specifically target proteins and peptides associated with this species. Only entries classified under the Reviewed (Swiss-Prot) section were selected for accuracy and reliability. To ensure bioavailability, proteins, and peptides with a maximum length of 200 amino acids were prioritized, with exceptions made for Fibroin (262 amino acids) and Sericin (1186 amino acids), as they are the primary components of *Bombyx mori*. The retrieved data underwent meticulous curation to eliminate duplicate entries, preserving the dataset’s integrity and accuracy. This step minimized redundancy and potential biases, enhancing the reliability and validity of the subsequent analyses. By rigorously collecting and curating data, a high-quality dataset of *Bombyx mori* bioactive constituents was established, forming a robust foundation for investigating their therapeutic potential in IHD. Detailed information about the employed computing power can be seen in Appendix A.

### 2.2. 3D Structure Modeling and MM2 Energy Minimization

The 3D structures of chemical compounds derived from *Bombyx mori* were generated using the Chem3D Ultra software version 22 (PerkinElmer, Waltham, MA, USA) with MM2 energy minimization applied to optimize molecular geometry and ensure structural stability [17,18]. For peptides and proteins, sequences were systematically collected from the UniProt database, targeting the *Bombyx mori* species. This process initially retrieved 27,941 records, of which 276 entries were classified under the Reviewed (Swiss-Prot) category. From these, 102 peptides and proteins with sequence lengths of 200 amino acids or fewer were selected to ensure favorable bioavailability. To maintain the quality and reliability of the dataset, rigorous selection and validation criteria were applied. Initially, sequence completeness was assessed to include only those with sufficient information for accurate 3D modeling, reducing the likelihood of incomplete or erroneous structural predictions [19]. This step was critical to ensuring high fidelity in downstream computational analyses. Quality control protocols were implemented to identify and rectify sequencing errors, ambiguous residues, or irregularities that could compromise the dataset’s integrity. Any sequences failing to meet these standards were excluded to ensure the robustness of subsequent analyses [20]. Duplicate entries were meticulously identified and removed to prevent redundancy and bias [21]. The final dataset was refined through a thorough comparison and validation process, ensuring that each unique peptide or protein sequence was represented only once. This curated dataset served as a reliable foundation for 3D structure generation and further computational studies, supporting a comprehensive exploration of the bioactive potential of *Bombyx mori* constituents.

The 3D structures of peptides and proteins derived from *Bombyx mori* were generated using AlphaFold version 3 (DeepMind Technologies Ltd., London, UK), a state-of-the-art deep learning platform renowned for its high accuracy in protein structure prediction [22]. AlphaFold excels in predicting protein structures even without homologous templates, making it particularly valuable for this study. Following structure generation, the active sites of the *Bombyx mori* peptides and proteins were analyzed using CASTpFold (University of Illinois at Chicago, IL, USA), a continuation of the widely recognized Computed Atlas of Surface Topography of Proteins (CASTp). This tool specializes in identifying and characterizing active sites and binding pockets within protein structures, providing crucial insights into their functional roles and potential interaction sites for ligands [23]. The target receptor for subsequent docking studies, identified through network pharmacology analysis, was retrieved from the Protein Data Bank (PDB) under accession code 6M0J (chain A) with a resolution of 2.45 Å [24]. This ensured the availability of a high-quality receptor structure for precise molecular docking simulations. The database of chemical compounds and protein sequences sourced from *Bombyx mori* is provided in Appendix A. For chemical compounds, the database includes PubChem CID, molecule name, molecular weight (MW), MlogP, hydrogen bond donors (HBD) and hydrogen bond acceptors (HBA), bioavailability score, blood–brain barrier (BBB) score, drug-likeness score, and canonical SMILES. For peptides and proteins, the database contains detailed information such as protein/peptide names, UniProt IDs, amino acid lengths, sequences, and identified binding site residues.

### 2.3. Network Pharmacology and Target Receptor Identification

The process of identifying potential target proteins for the bioactive compounds derived from *Bombyx mori* was conducted using a multi-step network pharmacology approach. Initially, bioactive compounds were screened based on two critical parameters: oral bioavailability (OB) and drug-likeness (DL). Compounds with OB ≥ 30% and DL ≥ 0.18 were retained for further analysis [25,26]. OB values were calculated using SwissADME (Molecular Modeling Group of the Swiss Institute of Bioinformatics, Lausanne, Switzerland) [27], while DL scores were derived using Molsoft (Molsoft LLC, La Jolla, CA, USA) [28]. This stringent filtering ensured that only compounds with favorable pharmacokinetic and drug-like properties were selected. To predict the potential protein targets of these bioactive compounds, the Similarity Ensemble Approach (SEA) was utilized. SEA uses the canonical Simplified Molecular Input Line Entry System (SMILES) of each compound, obtained from the PubChem database, to predict target proteins [29]. The prediction was limited to *Homo sapiens* proteins, with a Tanimoto Coefficient (TC) threshold set at ≥0.5. This threshold balances sensitivity and specificity, as higher thresholds reduce the number of predicted targets but increase the accuracy of associations [30]. Predicted target proteins from SEA were then merged, and duplicate entries were removed to create a non-redundant list. Next, target proteins associated with IHD were identified using the GeneCards database (https://www.genecards.org, accessed on 12 November 2024) [31]. A keyword search for “*ischemic heart disease*” yielded a large list of targets, which were filtered to include only those with a relevance score ≥ 10.00 to focus on the most relevant proteins. The resulting datasets of *Bombyx mori*-related and IHD-related target proteins were combined, and their intersection was determined to identify common targets potentially involved in therapeutic activity against IHD.

A component-target network was constructed to visualize and analyze these relationships using Cytoscape version 3.10.3 (Cytoscape Consortium, Washington, DC, USA) [32]. In this network, nodes represented bioactive compounds or target proteins, while edges indicated interactions between them. A common-target network was created by intersecting *Bombyx mori*-related and IHD-related target proteins. Key proteins within this network were identified based on their degree, defined as the number of direct connections to other nodes. Nodes with degrees greater than or equal to the median degree were considered pivotal for further analysis. To gain deeper insights into protein–protein interactions (PPIs), the stringApp plugin [33,34] in Cytoscape was employed. PPI networks for both *Bombyx mori*-related and IHD-related target proteins were constructed using *Homo sapiens* as the organism and a medium confidence score threshold of 0.400. These networks were then merged to identify intersecting proteins, representing critical nodes of interaction. The merged PPI network was analyzed using CytoNCA, a Cytoscape plugin for network analysis, to evaluate centrality measures such as degree centrality (DC), eigenvector centrality (EC), betweenness centrality (BC), and closeness centrality (CC). Only proteins meeting or exceeding the median values of these metrics were retained, as they were considered highly influential within the network. This comprehensive methodology identified bioactive compounds, their potential targets, and key proteins associated with IHD, providing a robust foundation for subsequent docking simulations.

### 2.4. Molecular Docking Simulation

This section provides a comprehensive analysis of the interactions between bioactive compounds derived from *Bombyx mori* and the target receptor, employing both ligand–protein and protein–protein docking simulations. The goal was to uncover critical details about the binding mechanism, including identifying key residues essential for ligand–protein and protein–protein complex formation. Additionally, we sought to gain insights into the types of intermolecular interactions, assess binding affinities, and explore the binding modes and orientations of the compounds at the receptor sites. To begin with, we used PDBSum (European Bioinformatics Institute, Cambridge, UK) [35], a computational tool designed to generate detailed summaries of protein structures and their interactions to delineate the binding sites of the target receptor. This analysis provided valuable information on the spatial arrangement of critical residues within the target protein’s binding site and their potential interactions with chemical compounds and proteins derived from *Bombyx mori*. Ensuring the accuracy of our docking analysis, we refined the target receptor’s structure using Swiss-PdbViewer version 4.1.1 (Swiss Institute of Bioinformatics, Lausanne, Switzerland) [36], a widely used tool for protein structure visualization and refinement, before proceeding with docking simulations. This refinement step ensured that the target receptor’s structure was optimal for further interaction studies. To assess the potential antagonistic effects of *Bombyx mori*-derived bioactive compounds on the target receptor, we compared them with well-established standard molecules known to interact with similar targets. Captopril was selected as the reference inhibitor ligand for the chemical compounds due to its well-documented role in inhibiting key cardiovascular receptors [37]. For the protein-based compounds, the DX600 peptide (sequence: GDYSHCSPLRYYPWWKCTYPDPEGGG) was used as the standard due to its known interactions with IHD-related receptors like ACE2 (IC_50_: 10.1 μM), as well as its therapeutic relevance in cardiovascular conditions [38]. The 3D structure of the DX600 peptide was generated using AlphaFold, ensuring a high-accuracy model for subsequent docking analysis.

Ligand–protein and protein–protein docking calculations were then performed using the High Ambiguity Driven Protein–Protein Docking (HADDOCK) platform version 2.4 (University of Utrecht, Utrecht, The Netherlands). HADDOCK is a well-established tool in computational biology designed to model ligand–protein and protein–protein interactions using ambiguous interaction restraints derived from experimental or computational sources [39,40]. The docking simulations were conducted within the advanced interface of the standalone version of HADDOCK, which allows for incorporating both geometric and energetic restraints to enhance the reliability of the results. The selection of optimal docking results for each ligand–protein and protein–protein complex was based on two critical criteria: the highest number of observed clusters or populations, which indicates the reliability and reproducibility of the predicted interactions, and the highest HADDOCK score, which reflects the strength of the binding affinity between the bioactive compounds from *Bombyx mori* and the target receptor. These criteria ensured that the most accurate and robust interaction models were chosen for further analysis, providing a solid foundation for understanding the potential therapeutic effects of the compounds. To further refine our binding energy predictions, we employed PRODIGY, a widely used computational tool for estimating binding free energy (ΔG) in kcal/mol. PRODIGY calculates the binding affinity based on structural and energetic features of ligand–protein and protein–protein complexes, incorporating factors such as intermolecular contacts and desolvation effects [41]. This approach enhances the reliability of our energy predictions by considering physiologically relevant conditions, providing a more accurate assessment of the potential stability and affinity of the docked complexes.

### 2.5. Molecular Dynamics (MD) Simulation

Molecular dynamics (MD) simulations were employed to examine the dynamic behavior and stability of ligand–protein and protein–protein complexes. These simulations were carried out using GROMACS 2023.3, a widely used molecular dynamics simulation software known for its precision and efficiency in modeling biomolecular systems [42]. The MD simulations enabled the investigation of the temporal evolution, structural stability, and conformational changes within these complexes. For the simulation of chemical compounds, the General Amber Force Field (GAFF2) was utilized to model the ligands in conjunction with partial charges derived from the Austin Model 1 semiempirical molecular orbital technique with Bond Charge Corrections (AM1-BCC). The *acpype* program was employed to assign the appropriate force field and partial charges to the ligands, using AmberTools21 for further processing. The simulation system was enclosed in a dodecahedron-shaped box, with a 1.4 nm distance between the protein’s largest principal radius and the edge of the box. Periodic boundary conditions were applied in all directions to replicate a bulk environment. The protein component of the complex was modeled using the AMBER99sb force field, and the system was solvated using SPC water molecules. To mimic physiological conditions, 100 mM NaCl was added to the system, along with counterions to neutralize the charge. For nonbonded interactions, a cutoff of 1.2 nm was used for short-range interactions, while long-range electrostatics were computed using the Particle Mesh Ewald (PME) method. The system underwent an initial energy minimization using the steepest descent method until the maximum force was reduced to below 1000 kJ/mol/nm. This was followed by a 100 ps restrained number of particles, volume, and temperature (NVT) equilibration, where the temperature was maintained at 310 K using the Berendsen thermostat. Using the Berendsen barostat, a 100 ps restrained number of particles, pressure, and temperature (NPT) equilibration was then performed to maintain pressure at 1 bar. After equilibration, a 100 ns unconstrained production simulation was conducted, with temperature and pressure maintained using the Berendsen thermostat and Parrinello–Rahman barostat, respectively.

In the case of protein-based compounds, the Optimized Potentials for Liquid Simulations (OPLS-AA/L) force field was applied to accurately model the molecular interactions within the protein–protein complexes [43]. The simulation box was configured using default cubic box parameters to ensure that the biomolecular complex was fully accommodated. The system was solvated with water molecules modeled using the Single Point Charge Extended (SPCE) model, and counterions were added to maintain charge neutrality [44]. The system underwent energy minimization using the steepest descent method to eliminate steric clashes and achieve a stable initial configuration. Following minimization, a two-phase equilibration process was applied. In the first phase, equilibration in the NVT ensemble was performed to regulate temperature and stabilize the system. In the second phase, equilibration was conducted in the NPT ensemble to maintain both constant pressure and temperature. Once the system reached equilibration, a production MD simulation was carried out for 100 nanoseconds to observe the long-term behavior and dynamics of the protein–protein complexes. During the MD simulations, several important parameters were monitored, including Root Mean Square Deviation (RMSD), Root Mean Square Fluctuation (RMSF), Radius of Gyration (RoG), and intermolecular hydrogen bonding interactions. These metrics provided valuable insights into the stability and conformational dynamics of the protein–protein interactions over time. To visualize key residues and intermolecular interactions, molecular visualization software such as PyMOL version 3.1.3 (Schrödinger LLC, New York, NY, USA) [45], BIOVIA Discovery Studio version 2024 (BIOVIA Dassault Systèmes, San Diego, CA, USA) [46], and UCSF Chimera version 1.18 (UCSF, San Francisco, CA, USA) [47] was used for manual inspection.

### 2.6. Molecular Mechanics/Poisson–Boltzmann Surface Area (MM/PBSA) Calculations

The Molecular Mechanics/Poisson–Boltzmann Surface Area (MM/PBSA) approach, coupled with MD simulations, was employed to evaluate the interaction between bioactive compounds (both chemical and protein-based) derived from *Bombyx mori* and the target receptor. MD simulations generated a variety of protein conformations, from which representative snapshots were selected for further analysis. Each selected snapshot underwent detailed energy calculations, which included gas-phase energy determination, solvation energy estimation using a continuum solvent model, and entropy evaluation [21,48]. To estimate the solvation energy, the solvent-accessible surface area (SASA) model was used to compute the nonpolar component of solvation energy. This model assumes that nonpolar solvation energy is proportional to the solvent-accessible surface area of the ligand–receptor complex, providing a more accurate representation of hydrophobic contributions to the binding energy. The Single-Trajectory Protocol (STP) was adopted for MM/PBSA calculations. This method assumes that the receptor and ligand undergo minimal conformational changes upon binding, allowing their free energies to be derived from the same trajectory. To perform these calculations, the *gmx_MMPBSA* module within the GROMACS simulation package was used [49]. The MM/PBSA binding free energy calculation is given by the following equation:
Δ*G_binding* = Δ*G_complex* − Δ*G_ligand/protein* − Δ*G_receptor*
where:

Δ*G_binding*: the binding free energy associated with forming the protein–protein complex.

Δ*G_complex*: the free energy of the fully solvated ligand–receptor protein–receptor complex.

Δ*G_ligand/protein*: the free energy of ligand/protein in its solvated state when unbound.

Δ*G_receptor*: the free energy of the receptor in its solvated state when unbound.

### 2.7. Pharmacophore Modeling and In Silico Toxicity Assessment

Pharmacophore modeling was conducted using LigandScout version 4.5 (Inte:Ligand, Vienna, Austria) [50] to identify and characterize the essential features required for the binding of chemical compounds derived from *Bombyx mori* to their target receptors. LigandScout 4.5 is a powerful tool for creating 3D pharmacophore models based on the spatial arrangement of key chemical features such as hydrogen bond donors/acceptors, hydrophobic regions, and electrostatic interactions, which are crucial for ligand–receptor interactions. This modeling approach helps identify potential bioactive compounds by highlighting critical pharmacophoric features essential for achieving effective binding affinity with the target proteins. In parallel, an in silico toxicity assessment was performed using DataWarrior version 6.4.1 (OpenMolecules, Karlsruhe, Germany), a computational tool designed to predict the toxicity profiles of chemical compounds [51]. DataWarrior integrates various molecular descriptors and toxicity prediction algorithms to evaluate the safety profile of the chemical compounds derived from *Bombyx mori*, ensuring that only those compounds with favorable toxicity profiles are considered for further biological studies.

## 3. Results

### 3.1. Identification of Drug–Target Interactions and Determination of Target Receptor

The network pharmacology analysis aimed to identify potential drug–target interactions and determine the most suitable target receptor for the bioactive compounds derived from *Bombyx mori*. A comprehensive literature review yielded 53 chemical compounds from *Bombyx mori*, which were initially filtered based on their pharmacological properties, specifically selecting compounds with an OB of ≥30% and a DL score of ≥0.18. Following this screening procedure, 17 chemical compounds were shortlisted for further investigation. To explore the possible interactions between these compounds and their target receptors, we employed the SEA platform, which enables the prediction of target proteins based on the chemical structure of each compound. After inputting the SMILES representations of the selected compounds into the SEA platform, we performed network analysis to generate a compound-target network. The resulting *Bombyx mori* component-target network consisted of 122 nodes and 935 edges. The nodes were categorized into two types: target proteins and bioactive compounds. The target proteins, shown as orange nodes, represented potential biological targets that could interact with the compounds. On the other hand, the chemical compounds derived from *Bombyx mori* were depicted as nodes with a green-to-blue gradient to illustrate their level of interaction. The darker the color, the greater their interaction with the target protein. This color gradient effectively visualizes the varying strengths of the compound–protein interactions, with darker shades representing a higher degree of interaction. Among the compounds analyzed, quercetin (PubChem CID: 5280343), luteolin (PubChem CID: 5280445), and apigenin (PubChem CID: 5280443) emerged as the top compounds. Each was associated with a high number of target proteins, suggesting their potential as key bioactive molecules for further exploration (Figure 1).

Among the identified cardiovascular-related targets, several bioactive compounds stood out for their potential cardiovascular benefits. Notably, quercetin (PubChem CID: 5280343) emerged as a compound with a broad spectrum of interactions. It was found to interact with several key target proteins, including Nitric Oxide Synthase 3 (NOS3, UniProt ID: P29474) and Cyclooxygenase 2 (COX2, UniProt ID: P35354), both of which are involved in regulating vascular tone, endothelial function, and the inflammatory response [52,53]. Quercetin’s ability to modulate these targets suggests its potential in improving endothelial health, reducing vascular inflammation, and enhancing nitric oxide production, all of which are critical for maintaining cardiovascular homeostasis. Another significant compound identified was Luteolin (PubChem CID: 5280444), which demonstrated interactions with several important cardiovascular-related proteins. Luteolin was found to target Adenosine A2A Receptor (ADORA2A, UniProt ID: P29274) and Matrix Metalloproteinase 9 (MMP9, UniProt ID: P14780). The adenosine receptors are known to play a role in regulating heart rate, vascular tone, and myocardial ischemia [54]. At the same time, matrix metalloproteinases are involved in tissue remodeling, which is crucial during the repair of vascular damage [55]. The interaction of luteolin with these targets suggests it may have potential therapeutic effects in reducing oxidative stress, improving blood flow, and preventing excessive tissue remodeling in cardiovascular diseases. Apigenin (PubChem CID: 5280443), another flavonoid compound, was also identified as a significant modulator of cardiovascular-related proteins. Apigenin was shown to interact with Carbonic Anhydrase 1 (CA1, UniProt ID: P00915) and Carbonic Anhydrase 2 (CA2, UniProt ID: P00918), both of which are involved in regulating blood pH and maintaining vascular function. Carbonic anhydrases play a role in modulating vascular tone [56], and their inhibition by apigenin could potentially help control blood pressure and prevent vascular spasms. Furthermore, apigenin’s interaction with these enzymes suggests it may have a role in improving endothelial function and promoting vascular relaxation. The complete results of the *Bombyx mori*-target network and interactions can be seen in Appendix A.

After identifying the potential target proteins associated with *Bombyx mori*-derived chemical compounds, we performed a STRING-based PPIs network analysis to investigate the relationships between these target proteins and known proteins involved in IHD. The analysis revealed that Angiotensin-Converting Enzyme 2 (ACE2) was the most prominent target receptor, showing extensive interactions with several cardiovascular-related proteins that are crucial in regulating vascular function and blood pressure (Figure 2). The PPI network highlighted ACE2’s involvement in various cardiovascular pathways. ACE2 plays a pivotal role in the renin–angiotensin system, regulating blood pressure and vascular tone by converting Angiotensin II into Angiotensin (1–7), which has vasodilatory properties [57]. Notably, ACE2 was shown to interact with MMP9, a protein involved in tissue remodeling and the degradation of extracellular matrix components. MMP9 plays a significant role in vascular inflammation and plaque instability, making it a critical factor in cardiovascular disease, especially in atherosclerosis [58]. The interaction between ACE2 and MMP9 suggests a potential therapeutic pathway for reducing vascular remodeling and inflammation in cardiovascular conditions. Furthermore, ACE2 was found to interact with Adenosine A2A Receptor (ADORA2A), a receptor involved in regulating vascular tone and myocardial ischemia. Adenosine signaling through ADORA2A can influence coronary blood flow and reduce oxidative stress, both of which are essential for maintaining cardiac health [59,60]. The binding between ACE2 and ADORA2A points to a possible mechanism where ACE2 modulates adenosine-mediated vasodilation, which could have protective effects on the cardiovascular system under conditions of stress or ischemia. In addition to its interactions with MMP9 and ADORA2A, ACE2 also showed connections with Apolipoprotein E (APOE), a protein that plays a central role in lipid metabolism and atherosclerosis. APOE is involved in the clearance of lipoproteins and the regulation of cholesterol homeostasis, processes that are closely linked to cardiovascular disease progression [61,62]. The interaction between ACE2 and APOE suggests that these proteins may collaborate to modulate lipid metabolism and vascular inflammation, potentially influencing the development of atherosclerosis. The network also revealed interactions between ACE2 and Nitric Oxide Synthase 3 (NOS3), a key enzyme in producing nitric oxide (NO), a molecule crucial for endothelial function and vasodilation [63,64]. The regulation of NO production could have significant implications for maintaining blood pressure and preventing endothelial dysfunction, both of which are central to the pathophysiology of cardiovascular diseases [65]. Thus, the STRING-based PPI network analysis underscored ACE2 as a central target receptor in the IHD-related protein network. Given its prominent position in this network, ACE2 was selected for further investigation through molecular docking and MD simulations to explore potential interactions with *Bombyx mori*-derived bioactive compounds.

### 3.2. Analysis of Molecular Interactions Through Molecular Docking Simulations

The molecular docking study provided insights into the binding affinities and interaction profiles of the top 10 chemical compounds derived from *Bombyx mori* against the ACE2 receptor, compared to the standard inhibitor, captopril. The results, summarized in Table 1, indicate that menaquinone-7, quercetin, and behenic acid outperformed other compounds based on their HADDOCK scores, binding energies, and interaction types. These findings highlight their potential as ACE2 inhibitors, with implications for therapeutic development. Among the tested compounds, menaquinone-7 demonstrated the strongest binding affinity with a HADDOCK score of −42.0 ± 3.0 and a binding energy of −10.09 kcal/mol, significantly surpassing captopril’s binding energy of −6.25 kcal/mol. The enhanced affinity of menaquinone-7 can be attributed to its hydrogen bond formation with Lys68, van der Waals interactions with Glu35 and Asp38, and alkyl interactions with Tyr83 and Pro84. These interactions, particularly the alkyl bonds, may provide additional stabilization within ACE2’s active pocket, making menaquinone-7 a promising candidate for ACE2 inhibition. Quercetin, the second top performer, exhibited a HADDOCK score of −20.9 ± 0.2 and a binding energy of −8.75 kcal/mol. Unlike menaquinone-7, quercetin formed a favorable network of hydrogen bonds, engaging Lys31, Asp38, and Lys353. Additionally, it established van der Waals interactions with Glu37 and His34, residues also targeted by captopril. This broader interaction profile, coupled with quercetin’s ability to form multiple hydrogen bonds, underscores its strong binding capability and potential as an ACE2 inhibitor. Behenic acid, ranking third, achieved a HADDOCK score of −25.6 ± 1.7 and a binding energy of −8.47 kcal/mol. Its interaction pattern was predominantly based on van der Waals forces, engaging residues such as Gln24, Phe28, Glu35, Gln76, Leu79, and Tyr83. While the lack of hydrogen bonds might limit its binding strength compared to menaquinone-7 and quercetin, its favorable van der Waals interactions indicate effective exploitation of the hydrophobic regions within ACE2’s active site. These findings align with previous experimental studies, which demonstrated that quercetin and its metabolites effectively inhibited recombinant human ACE2 activity with an IC_50_ of 4.48 μM [66]. Furthermore, another finding identified rutin and β-sitosterol as potent ACE2 binders through computational and bioinformatics screening, reinforcing the significance of plant-derived polyphenols in targeting ACE2 [67]. The consistency of our docking results with previously reported experimental data further substantiates the potential of quercetin as a promising natural inhibitor of ACE2, with implications for therapeutic applications against cardiovascular diseases and viral infections. The complete molecular docking results for chemical and protein-based compounds can be seen in Appendix A.

The molecular interactions of these top compounds with ACE2 were visualized in Figure 3 (3D binding poses) and detailed further in Figure 4 (2D interaction maps). Each compound displayed distinct interaction profiles while sharing key similarities with captopril, the standard inhibitor. Captopril forms hydrogen bonds with Glu35, van der Waals interactions with His34, Glu37, and Asp38, and an attractive charge interaction with Lys353. This well-characterized interaction network underpins its inhibitory activity. Menaquinone-7 mirrored some of these interactions, such as van der Waals forces with Glu35 and Asp38, while introducing additional stabilization through alkyl interactions with Tyr83 and Pro84. Quercetin extended the interaction network further, forming multiple hydrogen bonds and interacting with residues like Lys353, Glu37, and His34, which are also targeted by captopril. Behenic acid, despite relying solely on van der Waals forces, managed to interact with several hydrophobic and polar residues, including Gln24, Glu35, and Tyr83, demonstrating its adaptability within the ACE2 active pocket. The distinct binding profiles of menaquinone-7, quercetin, and behenic acid suggest varying mechanisms of ACE2 inhibition. Menaquinone-7’s strong binding affinity and stabilizing interactions indicate its suitability as a lead compound for further optimization. Quercetin’s ability to form multiple hydrogen bonds highlights its potential for high specificity and stability in binding, while behenic acid’s favorable van der Waals interactions make it a viable candidate for targeting hydrophobic regions within ACE2.

The molecular docking simulations also evaluated the interaction of *Bombyx mori*-derived proteins with the ACE2 receptor, revealing their potential as inhibitors by comparing their binding efficiency to the standard inhibitor DX600 peptide. The results, as summarized in Table 2, highlight diverse interaction profiles for the top 10 performing protein–receptor complexes, emphasizing variations in binding energies, van der Waals forces, electrostatic interactions, and desolvation energies. Among the candidates, Bombyxin A-5, Chorion class high-cysteine HCB protein 13, and FMRFamide-related peptides emerged as some of the most favorable protein-based inhibitors due to their strong binding affinities and interaction profiles. Bombyxin A-5 demonstrated the highest HADDOCK score of −113.2 ± 10.0 and a binding energy of −13.4 kcal/mol, indicating highly favorable interactions at the ACE2 receptor’s active site. Chorion class high-cysteine HCB protein 13 displayed a HADDOCK score of −71.4 ± 18.1 and a binding energy of −15.0 kcal/mol, showcasing robust molecular interactions driven by a combination of van der Waals and electrostatic forces. Similarly, FMRFamide-related peptides showed competitive binding with a HADDOCK score of −65.3 ± 12.9 and a binding energy of −12.8 kcal/mol. These peptides formed specific hydrogen bonds and exhibited balanced van der Waals and electrostatic contributions, further reinforcing their potential as ACE2 inhibitors. Other proteins, such as Chorion class B protein M1768 and NADH dehydrogenase 1 beta subunit 10, also exhibited competitive binding profiles. Chorion class B protein M1768 demonstrated a HADDOCK score of −89.6 ± 9.1 and a binding energy of −14.5 kcal/mol, with well-balanced van der Waals and electrostatic forces. NADH dehydrogenase 1 beta subunit 10 displayed a HADDOCK score of −81.3 ± 12.3 and a binding energy of −13.6 kcal/mol, emphasizing its strong electrostatic interactions and overall stability. Diuretic hormone 45 and Chorion class B protein L12 also emerged as noteworthy candidates, with HADDOCK scores of −98.9 ± 14.7 and −95.1 ± 4.4, respectively. Diuretic hormone 45 demonstrated significant hydrogen bonding and desolvation energy, indicative of stable binding. Chorion class B protein L12, on the other hand, showed well-balanced energy contributions and moderate hydrogen bonding.

Chorion class high-cysteine HCB protein 13 exhibited two hydrogen bonds. Cys34 formed a bond with Thr52 of ACE2, while Cys47 interacted with Gln42 of ACE2. These interactions, complemented by strong van der Waals and electrostatic contributions, make this protein a promising ACE2 inhibitor (Figure 5B). Similarly, Bombyxin A-5 formed two critical hydrogen bonds: one between Arg31 of Bombyxin A-5 and Thr20 of ACE2 and another between Pro22 of Bombyxin A-5 and Asp38 of ACE2. These hydrogen bonds were supplemented by van der Waals interactions, enhancing the stability and specificity of the complex. This robust interaction profile is illustrated in Figure 5C. FMRFamide-related peptides formed two hydrogen bonds, one between Arg34 and Ser19 of ACE2 and another between Ile43 and Lys26 of ACE2. These interactions, supported by van der Waals forces, underscored the peptide’s ability to engage the active site effectively (Figure 5D). The standard inhibitor DX600 peptide, while showing a reasonable HADDOCK score of −76.8 ± 0.9 and a binding energy of −8.6 kcal/mol, underperformed compared to the *Bombyx mori*-derived proteins. DX600 formed only a single hydrogen bond between Trp15 of DX600 and Thr27 of ACE2 (Figure 5A). This interaction, though stable, lacked the complexity and multiplicity seen in the interactions of the other candidates.

Table 3 summarizes the intermolecular contacts (ICs) and non-interacting surface areas (NIS) of the top protein–receptor complexes, highlighting significant differences in the nature and extent of these interactions compared to the standard DX600 peptide. First, the DX600 peptide, which serves as the standard inhibitor, exhibited a total of 3 charged-charged, 3 charged-polar, 14 charged-apolar, 0 polar-polar, 5 polar-apolar, and 6 apolar-apolar intermolecular contacts. The total NIS for the DX600 peptide was 27.88 Å^2^ for the charged residues and 33.63 Å^2^ for the apolar residues. These results reflect the relatively moderate level of interaction, with the charged-apolar contacts being the most prevalent. The peptide’s surface area analysis also showed a moderate degree of interaction with the ACE2 receptor. Among the top-performing proteins, Chorion class high-cysteine HCB protein 13 exhibited the most complex interaction profile, with a total of 6 charged-charged, 18 charged-polar, 26 charged-apolar, 5 polar-polar, 32 polar-apolar, and 13 apolar-apolar intermolecular contacts. Its total NIS values were 22.20 Å^2^ (charged) and 39.23 Å^2^ (apolar). This protein exhibited the highest number of charged-polar and polar-apolar interactions, contributing to its strong binding affinity with the ACE2 receptor, making it a favorable candidate for inhibition. Bombyxin A-5 also demonstrated strong interaction characteristics, with 3 charged-charged, 6 charged-polar, 25 charged-apolar, 0 polar-polar, 22 polar-apolar, and 18 apolar-apolar contacts. Its NIS values of 26.73 Å^2^ (charged) and 35.25 Å^2^ (apolar) reflect a good balance of interactions and minimal non-interacting surface area. Despite the relatively lower number of charged-charged and polar-polar interactions, its high charged-apolar interactions contributed to a favorable binding profile with ACE2. Other proteins, such as Chorion class B protein M1768 and FMRFamide-related peptides, also exhibited favorable interaction patterns with ACE2. Chorion class B protein M1768 showed 2 charged-charged, 6 charged-polar, 36 charged-apolar, 1 polar-polar, 26 polar-apolar, and 19 apolar-apolar contacts. Its NIS values were 23.96 Å^2^ (charged) and 40.83 Å^2^ (apolar). Meanwhile, FMRFamide-related peptides presented 9 charged-charged, 10 charged-polar, 27 charged-apolar, 2 polar-polar, 19 polar-apolar, and 14 apolar-apolar contacts, with NIS values of 28.45 Å^2^ (charged) and 35.61 Å^2^ (apolar). These results underscore their strong potential as ACE2 inhibitors due to their diverse intermolecular interactions. The complete molecular interactions of chemical and protein-based compound–receptor complexes can be seen in Appendix A.

The molecular docking simulations not only reveal the binding energies but also uncover the relationships between these energies and the various components contributing to the overall binding interaction. Figure 6A presents the binding affinity values (ΔG in kcal/mol) of the top 10 chemical compounds derived from *Bombyx mori*, showing their docking performance at the active sites of the ACE2 receptor. The compounds exhibited a range of binding affinities, with some demonstrating highly favorable interactions with ACE2. In Figure 6B, the correlation matrix for the chemical compounds derived from *Bombyx mori* provides a detailed analysis of the relationships between binding energy (ΔG in kcal/mol) and the individual energy components, including Van der Waals energy, electrostatic energy, and desolvation energy. The correlation between binding affinity and Van der Waals energy was found to be moderately positive (0.38), indicating that as the Van der Waals energy increases, the binding affinity tends to improve, albeit to a modest degree. The correlation with electrostatic energy was stronger (0.54), suggesting a significant contribution of electrostatic interactions to the binding process. The highest positive correlation was observed with desolvation energy (0.62), indicating that compounds with favorable desolvation energy are likely to exhibit stronger binding affinities to ACE2.

Figure 6C showcases the binding affinity values of the top 10 protein-based compounds derived from *Bombyx mori* docked to ACE2. These protein-based compounds demonstrated a range of binding affinities, with some achieving even stronger binding than the chemical compounds. The binding energy values varied, reflecting the diversity of interactions between the proteins and the ACE2 receptor. The data from this analysis indicate that protein-based compounds, particularly those with a higher number of intermolecular contacts, can exhibit more favorable binding affinities, underscoring their potential as ACE2 inhibitors. Figure 6D provides the correlation matrix for the protein-based compounds, illustrating the relationships between binding affinity (ΔG in kcal/mol) and the individual energy components. The correlation between binding affinity and Van der Waals energy was strongly positive (0.69), suggesting that the Van der Waals interactions play a crucial role in stabilizing the protein–ACE2 complex. Interestingly, the correlation between binding affinity and electrostatic energy was negative (−0.21), indicating that, for some protein-based compounds, the electrostatic interactions may not significantly contribute to the binding affinity or may even reduce it. The correlation with desolvation energy was positive (0.21), indicating that desolvation effects can still contribute to the binding affinity, although the correlation is weaker than for the chemical compounds.

### 3.3. Evaluation of Molecular Dynamics: Stability, Interactions, and Binding Affinity

The MD simulations for both chemical and protein-based compounds derived from *Bombyx mori* docked to the ACE2 were conducted to evaluate their stability, interaction patterns, and binding affinity (Table 4). These parameters provide critical insights into the dynamic nature of ligand–receptor interactions and the potential efficacy of these compounds as ACE2 inhibitors. RMSD is a widely used metric to assess the structural stability of the ligand–receptor complex during MD simulations. It reflects how much the ligand deviates from its initial position as the simulation progresses [68]. Lower RMSD values generally indicate a more stable interaction between the ligand and the receptor, while higher values suggest instability or significant conformational changes [69]. For the chemical compounds, the RMSD values ranged from 1.250 Å for Captopril_ACE2 (standard inhibitor) to 1.482 Å for Behenic acid_ACE2. Captopril_ACE2, the standard inhibitor, exhibited the lowest RMSD value, indicating a highly stable binding pose throughout the simulation. Other chemical compounds like Menaquinone-7_ACE2 (1.381 Å) and Quercetin_ACE2 (1.317 Å) also maintained relatively stable binding with ACE2, exhibiting RMSD values under 1.5 Å. This indicates that these compounds formed strong, consistent interactions with the ACE2 receptor. In contrast, compounds such as Behenic acid_ACE2 (1.482 Å) and Stearic acid_ACE2 (1.412 Å) demonstrated slightly higher RMSD values, suggesting that they may have undergone minor structural changes or experienced more flexibility in their binding pose during the simulation. In comparison, the protein-based compounds exhibited significantly higher RMSD values, ranging from 3.152 Å for DX600 peptide_ACE2 to 3.463 Å for Diuretic hormone 45_ACE2. These higher RMSD values suggest that, although the protein-based ligands interact with ACE2, they may be more flexible and undergo larger conformational changes during the simulation. The flexibility observed here may be a characteristic of protein–ligand interactions, where larger biomolecules can adapt their structure more readily in response to the dynamic nature of the receptor. At the end of the 100 ns MD simulation, there were no significant spikes in the RMSD values for either the chemical or protein-based compounds derived from *Bombyx mori* when interacting with ACE2. This suggests that the ligand–receptor complexes remained stable throughout the simulation, and the compounds did not exhibit sudden conformational changes or instability during the 100 ns timeframe. The absence of significant spikes further reinforces the reliability and consistency of the interactions, indicating that both types of compounds maintain steady binding poses with ACE2 over time.

Figure 7 presents the RMSF profiles of the Menaquinone-7_ACE2, Quercetin_ACE2, and Behenic acid_ACE2 complexes, alongside the Captopril_ACE2 complex. The RMSF profiles reveal notable similarities in the fluctuations of the binding regions for the chemical compounds. Specifically, significant fluctuations were observed in the Glu75-Thr92 and Asn330-Met360 regions of ACE2, which correspond to the binding sites where hydrogen bonds appear to break during the simulation. These fluctuations were particularly pronounced for Menaquinone-7_ACE2, Quercetin_ACE2, and Behenic acid_ACE2, even surpassing the fluctuations seen with the Captopril_ACE2 complex. This suggests that these three chemical compounds derived from *Bombyx mori* interact with ACE2 in a manner similar to Captopril, with a comparable ability to disrupt hydrogen bonds and induce conformational changes at key binding sites, indicating their potential to act as ACE2 inhibitors. Similarly, the protein-based compounds derived from *Bombyx mori*—namely, Chorion class high-cysteine HCB protein 13, Bombyxin A-5, and FMRFamide-related peptides—demonstrated RMSF profiles that were comparable to that of the DX600 peptide_ACE2 complex, the standard protein-based inhibitor. In these protein–ACE2 complexes, the RMSF values also indicated significant fluctuations in the Glu75-Thr92 and Asn330-Met360 regions of ACE2, with even higher fluctuations than those observed for the DX600 peptide_ACE2 complex. These higher fluctuations and the disruption of hydrogen bonds in these critical regions suggest that the three protein-based compounds exhibit similar inhibitory actions on ACE2, akin to the behavior of the DX600 peptide.

The RoG measures the compactness of the ligand–receptor complex, with smaller values indicating a more compact structure and larger values reflecting a more expanded or flexible complex. For the chemical compounds, the RoG values ranged from 2.185 Å for Captopril_ACE2 to 2.214 Å for Phytonadione_ACE2. These values were relatively consistent across the chemical compounds, suggesting that the ligand–receptor complexes remained fairly compact throughout the simulations, with no significant expansion or contraction. This suggests that these chemical compounds maintained stable, compact interactions with ACE2, contributing to their stability as potential inhibitors. The protein-based compounds, however, exhibited slightly higher RoG values, ranging from 2.794 Å for Chorion class B protein L12_ACE2 to 2.822 Å for Chorion class B protein M1768_ACE2. This indicates that the protein–ligand complexes were somewhat less compact than the chemical compound complexes, which is expected due to the larger and more flexible nature of proteins. However, the RoG values still remained within a relatively narrow range, indicating that despite their larger size, the protein-based ligands did not undergo dramatic conformational changes during the simulations.

The number of hydrogen bonds formed between the ligand and the ACE2 receptor is a key indicator of the strength and specificity of the ligand–receptor interaction. Hydrogen bonds play a crucial role in stabilizing the ligand–receptor complex and facilitating tight binding. Among the chemical compounds, the number of hydrogen bonds ranged from 3 for Behenic acid_ACE2 and Stearic acid_ACE2 to 6 for Rutin_ACE2. Rutin_ACE2 exhibited the highest number of hydrogen bonds, suggesting that it forms a particularly strong interaction with ACE2. Other chemical compounds, such as Menaquinone-7_ACE2 and Phytonadione_ACE2, formed 5 hydrogen bonds, indicating moderately strong interactions. On the other hand, compounds like Behenic acid_ACE2 and Palmitic acid_ACE2 formed only 3 hydrogen bonds, which may suggest weaker interactions with ACE2 compared to the other chemical compounds. For the protein-based compounds, the number of hydrogen bonds ranged from 8 for DX600 peptide_ACE2 to 13 for Chorion class high-cysteine HCB protein 13_ACE2. The protein-based ligands consistently formed more hydrogen bonds than the chemical compounds, with the highest values observed for Chorion class high-cysteine HCB protein 13_ACE2 (13 hydrogen bonds) and FMRFamide-related peptides_ACE2 (12 hydrogen bonds). These higher values reflect the greater number of interaction sites available on larger proteins, which can form multiple hydrogen bonds with the receptor and thus enhance the strength of the binding.

The MM/PBSA binding affinity calculations provide valuable insights into the interaction strength between chemical and protein-based compounds derived from *Bombyx mori* and the ACE2. For the chemical compounds, the ΔG_binding values ranged from −21.08 kcal/mol for Captopril_ACE2 (the standard inhibitor) to −35.12 kcal/mol for Menaquinone-7_ACE2. Menaquinone-7 exhibited the most favorable binding affinity among the chemical compounds, suggesting a strong interaction with ACE2. Other compounds, such as Quercetin_ACE2 (−29.98 kcal/mol), Behenic acid_ACE2 (−27.76 kcal/mol), and Stearic acid_ACE2 (−27.01 kcal/mol), also demonstrated notable binding affinities, though they were not as strong as Menaquinone-7. The calculated ΔG_binding values for these compounds indicate that they could form stable interactions with ACE2, potentially serving as effective inhibitors. On the other hand, compounds like Palmitic acid_ACE2 (−25.66 kcal/mol) and Luteolin_ACE2 (−25.89 kcal/mol) showed relatively weaker binding, suggesting a less stable interaction with ACE2.

In contrast, the protein-based compounds derived from *Bombyx mori* exhibited significantly stronger binding affinities compared to the chemical compounds. The ΔG_binding values for the protein-based compounds ranged from −140.36 kcal/mol for Chorion class CA protein ERA.5_ACE2 to −212.43 kcal/mol for Chorion class high-cysteine HCB protein 13_ACE2, with several proteins, such as Bombyxin A-5_ACE2 (−209.36 kcal/mol), FMRFamide-related peptides_ACE2 (−198.93 kcal/mol), and NADH dehydrogenase 1 beta subunit 10_ACE2 (−198.03 kcal/mol), also showing strong binding affinities. These proteins exhibited much more favorable ΔG_binding values than the chemical compounds, which suggests that the protein-based ligands derived from *Bombyx mori* might have higher efficacy in interacting with ACE2. The highest affinity was observed for the Chorion class high-cysteine HCB protein 13_ACE2, indicating that this protein could serve as a highly effective ACE2 inhibitor. The data presented in Table 5 highlights a clear trend where the protein-based compounds consistently exhibit stronger binding affinities than the chemical compounds. This difference is likely due to the larger size and more complex structure of the protein-based ligands, which allow them to form multiple interactions with the receptor, including hydrogen bonds, van der Waals forces, and electrostatic interactions. The results suggest that *Bombyx mori* proteins, particularly Chorion class high-cysteine HCB protein 13_ACE2, may serve as promising candidates for further investigation in drug development targeting ACE2, especially in comparison to smaller chemical inhibitors like Captopril.

### 3.4. Pharmacophore Modeling and Toxicity Profile Assessment

The pharmacophore modeling analysis of the chemical compounds derived from *Bombyx mori* revealed important insights into their potential interactions with ACE2, particularly in comparison to Captopril, the standard inhibitor. Among the top three performing ligands (Menaquinone-7, Quercetin, and Behenic acid), similar pharmacophore profiles were observed, particularly in their ability to generate hydrogen bond acceptors (HBAs) within the binding sites of ACE2. Captopril, the reference inhibitor, formed three HBAs through the hydroxyl groups in its structure alongside two hydrogen bond donors (HBDs) from its SH group (Figure 8A). This established a balanced network of interactions crucial for its binding affinity with ACE2. Menaquinone-7, on the other hand, predominantly exhibited hydrophobic interactions, primarily contributed by its isoprenoid side chain. In addition to these hydrophobic interactions, Menaquinone-7 also formed two hydrogen bond acceptors via its quinone ring, aligning with the pharmacophore features that favor binding to ACE2 (Figure 8B). Quercetin displayed the most abundant hydrogen bonding features among the top three compounds, with a total of six HBDs and three HBAs. These interactions were primarily facilitated by the hydroxyl groups on the benzene rings and the carbonyl groups present in the structure, providing multiple points of attachment to ACE2 (Figure 8C). This extensive hydrogen bonding ability suggests that Quercetin could form a more stable and potentially stronger interaction with ACE2 compared to Menaquinone-7 and Behenic acid. Behenic acid, while still showing some pharmacophore similarity to Captopril, had fewer hydrogen bonding interactions. It only formed one HBA, which was generated by its carboxyl group (Figure 8D). The limited hydrogen bonding potential of Behenic acid compared to the other two compounds suggests that its binding affinity might be weaker, although its hydrophobic interactions could still contribute to its overall binding to ACE2. These results indicate that the top three chemical compounds derived from *Bombyx mori* (Menaquinone-7, Quercetin, and Behenic acid) share similarities with Captopril in terms of their hydrogen bonding properties, with Quercetin showing the most abundant hydrogen bonding potential.

The drug-likeness and toxicity profiles of the *Bombyx mori*-derived chemical compounds targeting ACE2 were evaluated based on several parameters, including Lipinski’s Rule of Five violations, mutagenic, tumorigenic, reproductive, and irritant potential (Table 6). In terms of Lipinski’s Rule of Five, Menaquinone-7 exhibited two violations: its molecular weight (MW) exceeds 500 g/mol and its LogP value is greater than 5. These violations suggest that Menaquinone-7 may have lower oral bioavailability, but the compound still showed acceptable drug-likeness with a score of 0.62. Importantly, Menaquinone-7 was deemed non-mutagenic, non-tumorigenic, and showed no reproductive or irritant effects, indicating a favorable safety profile. Despite the Lipinski violations, Menaquinone-7 appears to be a promising compound for ACE2 targeting, with good pharmacological potential. Quercetin, another top-performing compound, did not violate any of Lipinski’s rules, making it an ideal candidate from a drug-likeness perspective. Its drug-likeness score was 0.52, which is relatively moderate. However, Quercetin was flagged as highly mutagenic and tumorigenic, suggesting that its potential use in therapeutic applications may require careful consideration of these adverse effects. Like Menaquinone-7, Quercetin showed no issues with reproductive toxicity or irritant effects, but its high mutagenic and tumorigenic potential warrants further investigation.

Behenic acid, with one Lipinski violation (LogP > 5), had a drug-likeness score of 0.54, indicating moderate drug-like properties. Behenic acid was found to be non-mutagenic, non-tumorigenic, and did not pose reproductive or irritant concerns. Although it has a higher LogP value, suggesting potential challenges with solubility and permeability, its relatively safe toxicity profile and moderate drug-likeness make it a potential lead compound for ACE2 inhibition. Other compounds, such as Stearic acid and Phytonadione, exhibited similar toxicity profiles to Behenic acid, with Stearic acid also being flagged for high mutagenic and tumorigenic potential and Phytonadione showing no toxicity concerns. Stearic acid violated no Lipinski rules, while Phytonadione violated one (LogP > 5), but both compounds still demonstrated favorable drug-likeness scores (0.54 and 0.93, respectively). Rutin, although showing three Lipinski violations (MW > 500 g/mol, HBA > 10, HBD > 5), demonstrated an excellent drug-likeness score of 0.91. Despite the significant violations, rutin showed no mutagenic, tumorigenic, reproductive, or irritant concerns, making it an interesting compound for further exploration in drug design. The high drug-likeness score of Rutin compensates for its Lipinski violations, suggesting that it may still be a viable candidate for ACE2-targeted therapy. Several other compounds, including Tocopherol, Eicosenoic acid, Luteolin, and Palmitic acid, displayed a range of Lipinski violations (mostly related to LogP values greater than 5), with varying drug-likeness scores. However, most of these compounds showed no mutagenic, tumorigenic, or reproductive toxicity, indicating relatively safe profiles. Notably, Luteolin had a low drug-likeness score of 0.38, suggesting that it may not be a strong candidate for development despite its lack of toxicity concerns.

## 4. Discussion

This study presents a comprehensive exploration of the pharmacological mechanisms underlying the potential therapeutic effects of *Bombyx mori* (Abresham) for IHD, using a combination of network pharmacology, molecular docking, MD simulations, and pharmacophore modeling. Among the top-performing chemical compounds, Menaquinone-7, Quercetin, and Behenic acid showed significant potential for interaction with ACE2, the prioritized receptor in this study. Additionally, protein-based compounds such as Chorion class high-cysteine HCB protein 13, Bombyxin A-5, and FMRFamide-related peptides demonstrated promising binding profiles, suggesting their potential roles in the therapeutic mechanisms of *Bombyx mori* for cardiovascular diseases. The pharmacophore modeling results revealed that Menaquinone-7, Quercetin, and Behenic acid share certain pharmacological features, particularly in their ability to form hydrogen bonds with ACE2. Menaquinone-7, with its hydrophobic isoprenoid side chain and quinone ring, exhibited a combination of hydrophobic and hydrogen bonding interactions. These interactions are critical for achieving stable binding to ACE2, which is an essential enzyme involved in cardiovascular regulation and inflammation. Quercetin, which displayed the highest number of hydrogen bonds, could form more stable interactions with ACE2, potentially enhancing its cardioprotective effects. Quercetin’s anti-inflammatory and antioxidative properties have been well documented in the context of cardiovascular diseases [70,71,72], and its role in ACE2 binding may further bolster its therapeutic potential [73,74]. In contrast, Behenic acid, with its limited hydrogen bonding capabilities, might exhibit weaker binding, but its hydrophobic interactions could still contribute to its overall efficacy in targeting ACE2.

In parallel to the chemical compounds, the protein-based bioactive molecules from *Bombyx mori*, namely Chorion class high-cysteine HCB protein 13, Bombyxin A-5, and FMRFamide-related peptides, were also subjected to molecular docking and MD simulations. Bombyxin A-5, a known neuropeptide from *Bombyx mori*, has been shown to possess anti-inflammatory properties [75,76], which are crucial for addressing the chronic inflammation often observed in IHD patients. The interaction of Bombyxin A-5 with ACE2 further suggests a mechanism by which *Bombyx mori* could modulate cardiovascular function through its neuropeptides. Similarly, Chorion class high-cysteine HCB protein 13, which is involved in cellular signaling and immune modulation [77], could contribute to the protective effects against IHD by interacting with key receptors in the cardiovascular system. The FMRFamide-related peptides, known for their involvement in modulating blood pressure and heart rate [78,79], could potentially enhance the therapeutic effects of *Bombyx mori* by targeting specific pathways associated with IHD pathophysiology. These protein-based compounds provide a unique mechanism of action, complementing the effects of chemical compounds. Furthermore, the toxicity profile analysis of the chemical compounds revealed that Menaquinone-7 and Behenic acid exhibited relatively safe profiles, despite some violations of Lipinski’s Rule of Five. Menaquinone-7’s higher molecular weight and LogP value suggest challenges related to oral bioavailability, but its favorable safety profile and pharmacological potential outweigh these concerns. Quercetin, despite showing no violations of Lipinski’s rules, exhibited high mutagenic and tumorigenic risks, highlighting the need for caution in its therapeutic application. This aligns with earlier studies that identified Quercetin’s cytotoxic potential at high concentrations, although its anti-inflammatory effects make it a viable candidate for cardiovascular therapies if carefully dosed [70,80]. In contrast, Behenic acid demonstrated a moderate drug-likeness score, making it a promising compound, especially in combination with other molecules to improve its therapeutic properties. The integration of these findings underscores the multifaceted therapeutic potential of *Bombyx mori* for IHD, bridging traditional Unani medicine with modern pharmacological research. By targeting ACE2, a key receptor involved in cardiovascular regulation, the chemical and protein-based compounds derived from *Bombyx mori* present an innovative approach to IHD therapy. The application of computational techniques, including molecular docking, MD simulations, and pharmacophore modeling, has provided a detailed understanding of how these bioactive compounds interact with IHD-related receptors, thus optimizing the therapeutic strategy.

## 5. Limitations, Clinical Implications, and Future Works

While this study provides valuable insights into the pharmacological mechanisms of *Bombyx mori* (Abresham) for IHD, there are several limitations that must be acknowledged. First, the in silico approach employed in this study, while powerful, is based on predictive models and simulations. These methods rely on the accuracy of the molecular docking, molecular dynamics simulations, and pharmacophore modeling algorithms, which may not fully capture the complexity of biological systems in vivo. Despite extensive validation through molecular dynamics simulations, the dynamic nature of receptor–ligand interactions and the potential for allosteric effects remain challenges that computational models may not fully address. Additionally, the lack of experimental validation in biological systems (e.g., cell-based assays or animal models) limits the generalizability of the findings to human clinical settings. The interactions between the bioactive compounds of *Bombyx mori* and other off-target molecules, which could lead to unexpected pharmacological effects, were not considered in this study. Another limitation is the reliance on computational toxicity predictions. While the molecular toxicity profiles of compounds like Menaquinone-7, Quercetin, and Behenic acid suggest favorable safety, these predictions are based on algorithms that may not fully reflect the complex interactions in a human body. In vitro and in vivo studies are essential to confirm the actual toxicity profiles of these compounds. Moreover, potential interactions between these compounds, whether additive, synergistic, or antagonistic, were not explored in depth. Furthermore, while the 100 ns molecular dynamics simulations provided significant insights into ligand–receptor stability, this duration may not be sufficient for exhaustive exploration of the energetic and conformational landscape of the systems. Advanced methods such as Replica Exchange Molecular Dynamics (REMD) or multiple independent short simulations with varied initial velocities could enhance sampling efficiency and improve convergence. However, due to computational resource constraints, these approaches were not feasible in the present study. Future studies could benefit from employing longer simulation times, additional replicas, or enhanced sampling techniques to further validate the stability and dynamic behavior of the identified bioactive compounds.

The findings of this study have significant implications for the clinical application of *Bombyx mori* as a therapeutic agent for IHD. The identification of Menaquinone-7, Quercetin, Behenic acid, Bombyxin A-5, Chorion class high-cysteine HCB protein 13, and FMRFamide-related peptides as potential bioactive molecules that target key receptors involved in IHD opens up new avenues for the development of novel cardiovascular drugs. These compounds, particularly Menaquinone-7 and Quercetin, are already recognized for their health benefits and are often available as dietary supplements. Incorporating *Bombyx mori* or its isolated bioactive compounds into clinical practice may offer additional cardioprotective benefits, particularly for patients with IHD who do not respond adequately to conventional therapies. Furthermore, the pharmacological effects of Bombyxin A-5 and other protein-based compounds suggest that *Bombyx mori* could be a source of natural peptides with therapeutic potential. Given that peptides often exhibit higher specificity and fewer side effects compared to traditional small molecules, they represent a promising class of drugs. The anti-inflammatory and immunomodulatory properties of Bombyxin A-5 and FMRFamide-related peptides could enhance the management of IHD, where chronic inflammation is a key driver of disease progression. The use of natural protein-based therapeutics, derived from a widely used traditional medicine, could offer a more accessible and cost-effective alternative to current treatment options.

Future studies should aim to address the limitations of this study by incorporating in vitro and in vivo experiments. These studies could involve cell culture models of cardiovascular disease, such as human endothelial cells or cardiomyocytes, to assess the efficacy and toxicity of the identified compounds at the molecular and cellular levels. Furthermore, animal models of IHD, particularly those that closely mimic human cardiovascular pathophysiology, would be essential for confirming the therapeutic potential of *Bombyx mori* extracts or isolated bioactive compounds in a living organism. This would also help determine the proper dosing regimens and identify any potential side effects. Another area for future work is the exploration of the synergistic effects of the compounds derived from *Bombyx mori*. Since traditional medicines often consist of multiple active ingredients, studying the interactions between different chemical and protein-based compounds could reveal enhanced therapeutic outcomes when used in combination. Investigating the molecular mechanisms of these compounds in greater detail through advanced techniques like RNA sequencing or proteomics could provide deeper insights into their effects on key signaling pathways associated with IHD. Additionally, the role of *Bombyx mori* in modulating cardiovascular function should be studied in the context of other diseases related to cardiovascular health, such as hypertension, heart failure, and atherosclerosis. Given the diversity of bioactive compounds in *Bombyx mori*, it is likely that its therapeutic applications extend beyond IHD alone, and a broader exploration of its pharmacological potential could yield valuable results. While this study utilized molecular docking and MM/PBSA calculations to predict binding affinity, experimental validation through binding assays would be essential for confirming the computational findings. Techniques such as surface plasmon resonance (SPR), isothermal titration calorimetry (ITC), or microscale thermophoresis (MST) could provide quantitative binding affinity measurements, validating the predicted interactions between *Bombyx mori* bioactive compounds and the target receptor. Furthermore, a critical limitation of this study is the inability to predict potential immunogenicity risks associated with *Bombyx mori*-derived proteins and peptides. Since these biomolecules originate from a non-human source, they may trigger immune responses, potentially limiting their clinical applicability. Future research should incorporate in silico immunogenicity prediction tools such as NetMHC, IEDB, or AlgPred to assess T-cell and B-cell epitope recognition. Additionally, in vitro immunogenicity assays, including cytokine release and dendritic cell activation studies, would provide valuable insights into the safety and biocompatibility of these bioactive compounds for therapeutic use.

To strengthen the computational framework, future studies could include the simulation of a negative control—a complex identified as unlikely to be a good candidate from the initial screening—to verify whether MM/PBSA binding free energy calculations correctly predict low binding affinity. Additionally, the Single-Trajectory Protocol (STP) was employed in this study for MM/PBSA calculations, assuming minimal conformational changes between the bound and unbound states of the receptor–ligand complex. While STP is generally suitable for small-molecule inhibitors and stable complexes, it may introduce limitations when applied to protein–protein interactions, where significant structural flexibility is expected. A Multiple-Trajectory Protocol (MTP), which accounts for independent conformational sampling of the receptor, ligand, and complex, could enhance accuracy in future studies, particularly for highly dynamic systems. Comparing STP and MTP approaches in future work would help refine binding free energy predictions and provide a more comprehensive understanding of biomolecular interactions. Furthermore, we acknowledge that captopril is primarily an ACE1 inhibitor rather than an ACE2-specific inhibitor, which may influence binding profiles due to differences in target selectivity. While previous studies have reported captopril’s ability to interact with ACE2, its affinity is notably lower compared to its interaction with ACE1. Given the structural similarities between ACE1 and ACE2 and the absence of a widely accepted ACE2-specific inhibitor for benchmarking, captopril was included as a comparative reference to provide context for the binding affinities of the tested compounds. However, future studies should consider incorporating more relevant ACE2-specific inhibitors, such as MLN-4760 [81], to improve the accuracy of comparative analyses.

We also recognize that our protein docking results did not explicitly account for potential post-translational modifications (PTMs) in *Bombyx mori* proteins, which could significantly influence binding interactions and inhibitor efficacy. PTMs such as phosphorylation, glycosylation, and disulfide bond formation can alter protein conformation and stability, potentially impacting the accuracy of our docking predictions. While structural models used in this study were based on available sequences and homology modeling, future research should integrate PTM predictions using bioinformatics tools such as ModPred, NetPhos, and GlycoEP to identify possible modification sites. Moreover, MD simulations incorporating PTM-mimicking modifications and experimental validation through mass spectrometry-based proteomics could provide deeper insights into PTM-induced structural changes. Investigating PTM effects on binding affinities will help refine computational models and enhance the reliability of docking analyses for *Bombyx mori*-derived bioactive compounds. Furthermore, metabolic stability plays a critical role in determining the pharmacokinetics and potential therapeutic viability of bioactive compounds. Our current study did not assess the metabolic stability of the identified compounds, which is essential for understanding their half-life, bioavailability, and potential metabolic degradation. Future studies should incorporate in silico metabolism predictions using tools like ADMET Predictor, MetaPrint2D, and SMARTCyp to estimate metabolic pathways and identify potential sites of biotransformation. Additionally, experimental validation through liver microsome stability assays and cytochrome P450 interaction studies would provide valuable insights into the metabolic fate of these compounds.

## 6. Conclusions

In conclusion, this study provides compelling evidence for the potential of *Bombyx mori* (Abresham) as a therapeutic agent for IHD, highlighting several bioactive compounds, including Menaquinone-7, Quercetin, Behenic acid, and protein-based molecules like Bombyxin A-5, Chorion class high-cysteine HCB protein 13, and FMRFamide-related peptides. Through an integrative molecular simulation approach, the study elucidates key interactions between these compounds and IHD-related receptors, suggesting their roles in modulating critical pathways such as inflammation, oxidative stress, and cellular apoptosis. The findings underscore the value of traditional Unani medicine, specifically *Bombyx mori*, as a promising source of novel cardiovascular therapeutics. However, further experimental validation in vitro and in vivo is necessary to confirm the clinical applicability and safety of these compounds, paving the way for future studies and potential integration into clinical practice.

## Figures and Tables

**Figure 1 pharmaceutics-17-00295-f001:**
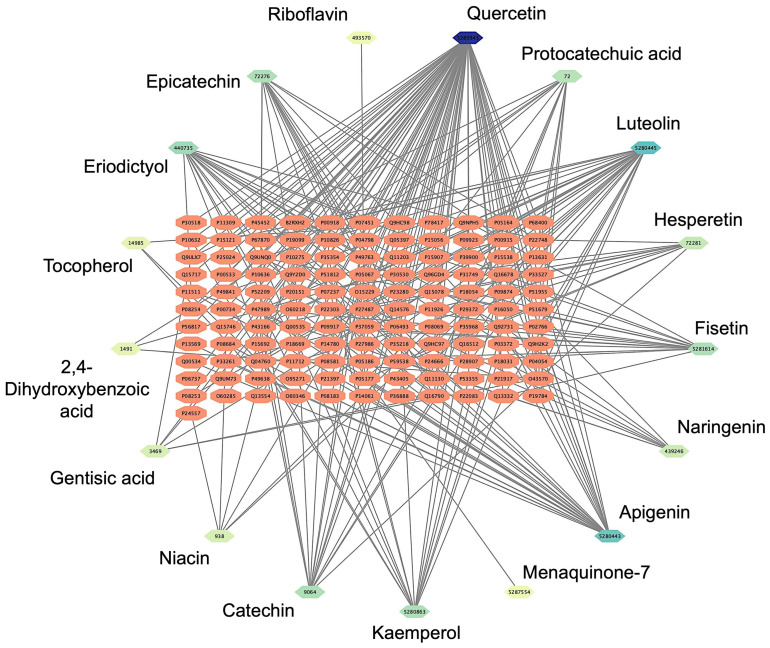
Results of the network pharmacology analysis. *Bombyx mori* compounds-target network with 140 nodes and 268 edges. Target proteins are depicted as orange nodes, while the chemical bioactive compounds derived from *Bombyx mori* are represented by green-to-blue gradient nodes.

**Figure 2 pharmaceutics-17-00295-f002:**
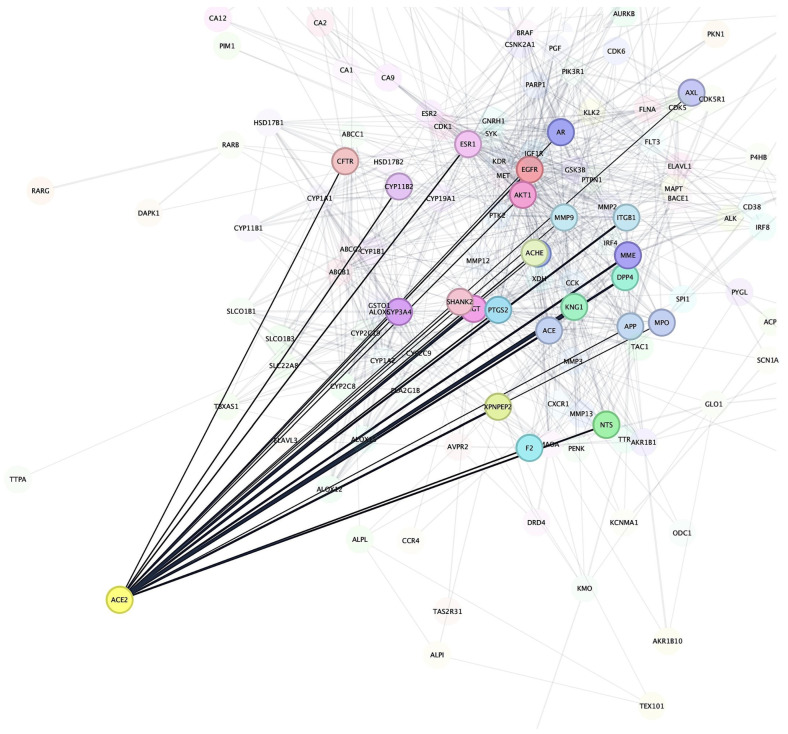
Results of the network pharmacology analysis. ACE2 is identified as the most prominent target receptor, showing extensive interactions with proteins targeted by the *Bombyx mori*-derived chemical compounds.

**Figure 3 pharmaceutics-17-00295-f003:**
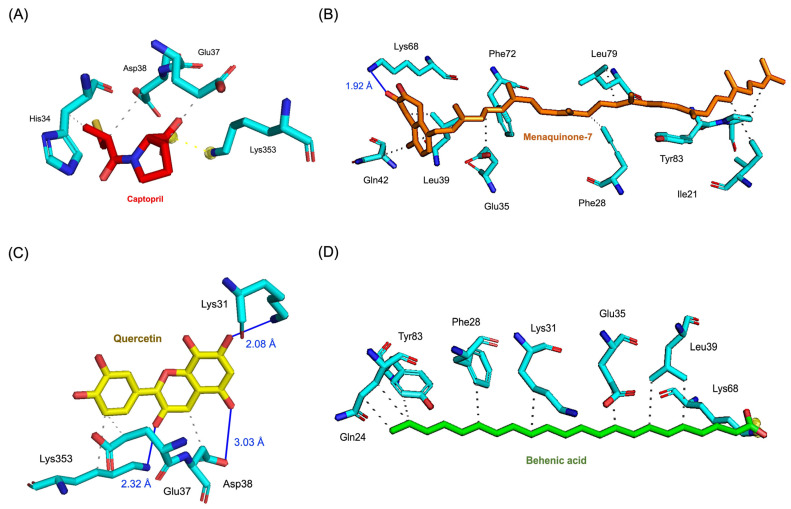
Results of molecular docking simulations. 3D representations of molecular poses and interactions between the ACE2 receptor and the top three performing compounds derived from *Bombyx mori*, alongside the standard inhibitor captopril. (**A**) Captopril_ACE2 complex, illustrating the reference binding interaction. (**B**) Menaquinone-7_ACE2 complex, showing its strong binding affinity and key interaction points. (**C**) Quercetin_ACE2 complex, highlighting its favorable hydrogen bonding network. (**D**) Behenic acid_ACE2 complex, demonstrating significant van der Waals interactions within the receptor’s active site.

**Figure 4 pharmaceutics-17-00295-f004:**
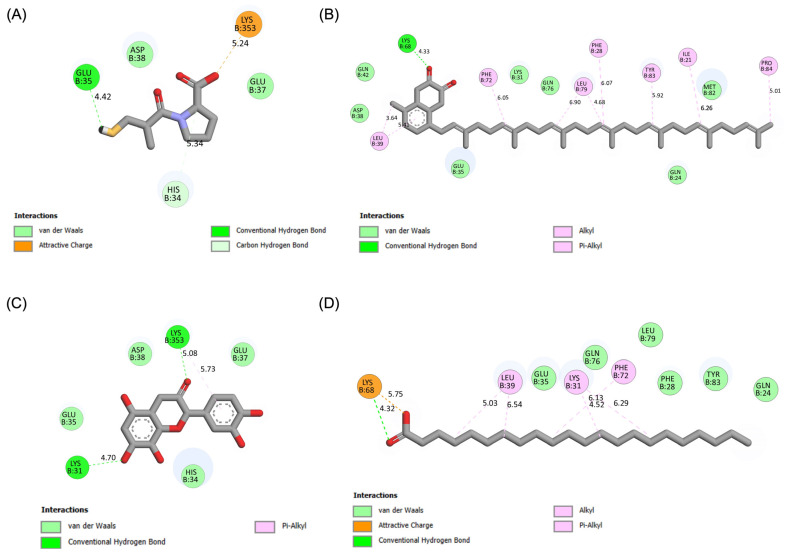
Results of molecular docking simulations. 2D interaction maps of the ACE2 receptor with the top three performing compounds derived from *Bombyx mori*, compared to the standard inhibitor captopril. (**A**) Captopril_ACE2 complex. (**B**) Menaquinone-7_ACE2 complex. (**C**) Quercetin_ACE2 complex. (**D**) Behenic acid_ACE2 complex.

**Figure 5 pharmaceutics-17-00295-f005:**
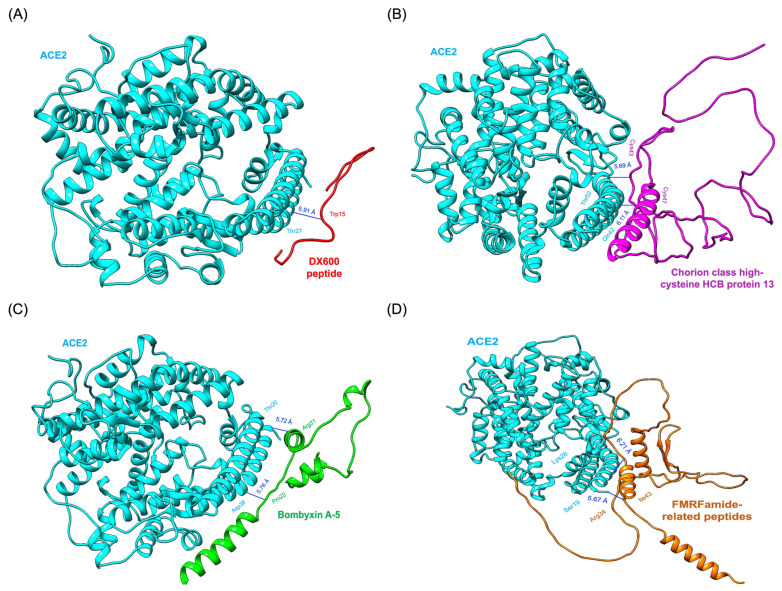
Results of molecular docking simulation. 3D binding poses of protein-based compounds derived from *Bombyx mori* interacting with the ACE2 receptor within the binding pocket. (**A**) DX600 peptide_ACE2 complex (standard inhibitor). (**B**) Chorion class high-cysteine HCB protein 13_ACE2 complex. (**C**) Bombyxin A-5_ACE2 complex. (**D**) FMRFamide-related peptides_ACE2 complex.

**Figure 6 pharmaceutics-17-00295-f006:**
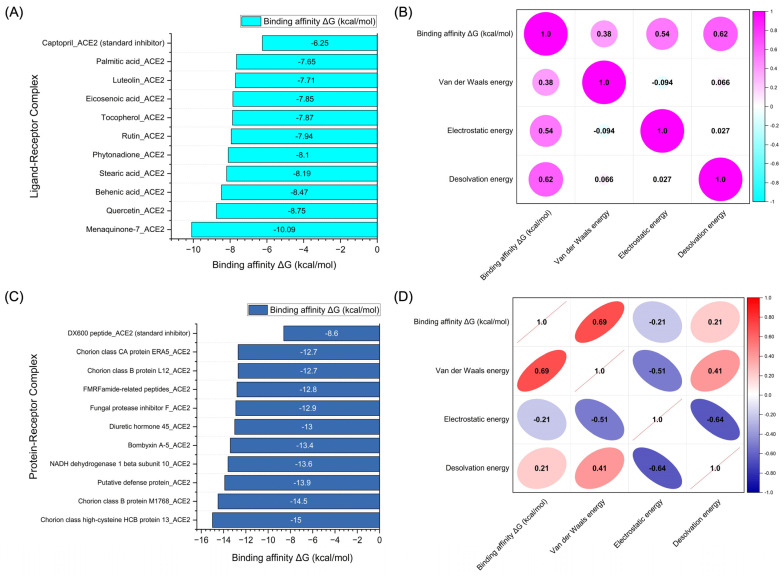
Results of molecular docking simulations. (**A**) Binding affinity values (kcal/mol) of the top 10 chemical compounds derived from *Bombyx mori* docked to the active sites of ACE2. (**B**) Correlation matrix depicting the relationship between binding energy (kcal/mol) and individual energy components for the chemical compounds derived from *Bombyx mori*. (**C**) Binding affinity values (kcal/mol) of the top 10 protein-based compounds derived from *Bombyx mori* docked to the active sites of ACE2. (**D**) Correlation matrix illustrating the relationship between binding energy (kcal/mol) and individual energy components for the protein-based compounds derived from *Bombyx mori*. The correlation values span from −1 to 1, with 1 representing a perfect positive correlation, −1 indicating a perfect negative correlation, and 0 signifying the absence of any correlation.

**Figure 7 pharmaceutics-17-00295-f007:**
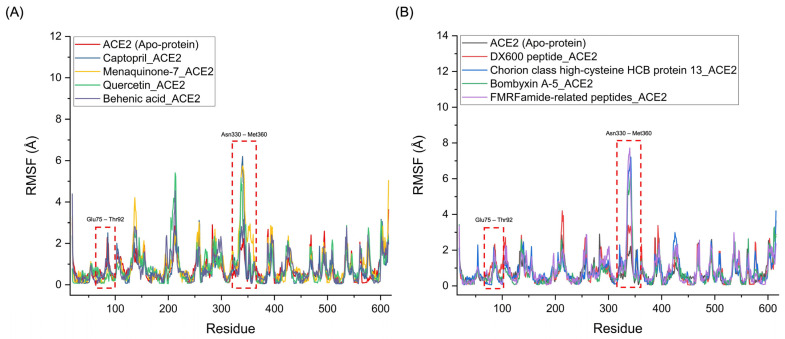
RMSF Profiles of ACE2-Ligand complexes. (**A**) RMSF profiles of chemical compounds derived from *Bombyx mori* (Menaquinone-7, Quercetin, and Behenic acid) compared to the standard inhibitor, Captopril, highlighting the fluctuations in key binding regions of ACE2. (**B**) RMSF profiles of protein-based compounds derived from *Bombyx mori* (Chorion class high-cysteine HCB protein 13, Bombyxin A-5, and FMRFamide-related peptides) compared to the standard protein inhibitor, DX600 peptide, showing similar fluctuations and disruption of hydrogen bonds in critical binding regions of ACE2.

**Figure 8 pharmaceutics-17-00295-f008:**
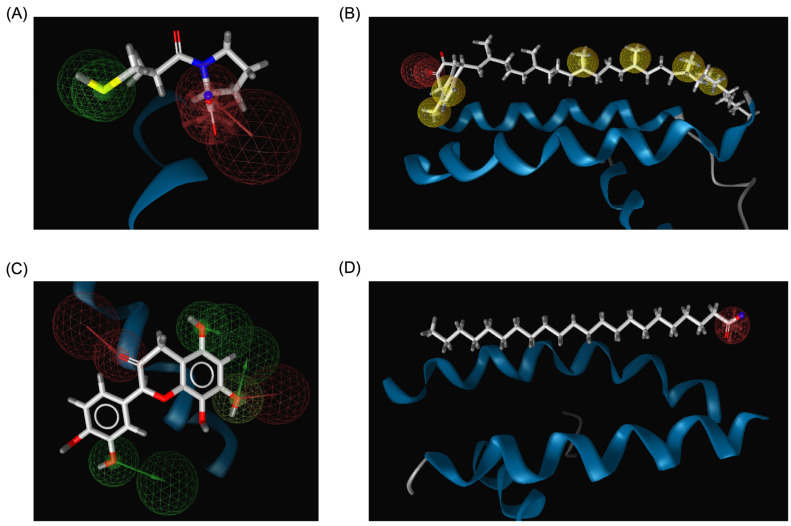
3D structure-based pharmacophore modeling results. (**A**) Pharmacophore model of the Captopril_ACE2 complex. (**B**) Pharmacophore model of the Menaquinone-7_ACE2 complex. (**C**) Pharmacophore model of the Quercetin_ACE2 complex. (**D**) Pharmacophore model of the Behenic acid_ACE2 complex. Hydrophobic interactions are depicted as yellow spheres, while hydrogen bond donors and hydrogen bond acceptors are represented by green and red arrows, respectively.

**Table 1 pharmaceutics-17-00295-t001:** Molecular docking results of top 10 performing *Bombyx mori*-derived chemical compounds in complex with ACE2 compared to the standard inhibitor (captopril).

Complex	HADDOCK Score (a.u.)	Binding Energy (kcal/mol)	Van der Waals Energy	Electrostatic Energy	Desolvation Energy	RMSD
Captopril_ACE2 (standard inhibitor)	−11.2 ± 1.8	−6.25	−12.6 ± 0.4	−40.5 ± 7.3	−1.5 ± 0.5	0.5 ± 0.0
Menaquinone-7_ACE2	−42.0 ± 3.0	−10.09	−24.1 ± 2.3	−88.3 ± 4.0	−9.4 ± 1.2	0.4 ± 0.0
Quercetin_ACE2	−20.9 ± 0.2	−8.75	−18.0 ± 0.2	−37.2 ± 2.2	0.7 ± 0.3	0.6 ± 0.0
Behenic acid_ACE2	−25.6 ± 1.7	−8.47	−14.2 ± 2.5	−94.2 ± 10.7	−6.4 ± 0.8	0.2 ± 0.0
Stearic acid_ACE2	−24.5 ± 3.3	−8.19	−13.7 ± 0.7	−88.2 ± 13.8	−5.0 ± 0.4	0.2 ± 0.0
Phytonadione_ACE2	−31.6 ± 1.2	−8.10	−24.6 ± 0.1	−43.3 ± 4.4	−3.6 ± 0.8	0.5 ± 0.0
Rutin_ACE2	−31.8 ± 0.1	−7.94	−26.1 ± 0.4	−22.6 ± 10.9	−3.4 ± 0.9	0.6 ± 0.0
Tocopherol_ACE2	−33.7 ± 1.4	−7.87	−26.1 ± 0.3	5.6 ± 0.2	−9.1 ± 0.3	0.1 ± 0.0
Eicosenoic acid_ACE2	−26.0 ± 3.5	−7.85	−17.1 ± 1.9	−99.1 ± 17.7	−2.4 ± 1.0	0.8 ± 0.0
Luteolin_ACE2	−19.5 ± 0.5	−7.71	−16.8 ± 0.4	−26.2 ± 2.3	−0.1 ± 0.3	0.9 ± 0.0
Palmitic acid_ACE2	−25.7 ± 0.7	−7.65	−18.4 ± 0.3	−61.6 ± 5.5	−5.8 ± 0.1	0.6 ± 0.0

**Table 2 pharmaceutics-17-00295-t002:** Molecular docking results of top 10 performing protein–receptor complexes compared to the standard inhibitor (DX600 peptide).

Complex	HADDOCK Score (a.u.)	Binding Energy (kcal/mol)	Van der Waals Energy	Electrostatic Energy	Desolvation Energy	RMSD
DX600 peptide_ACE2 (standard inhibitor)	−76.8 ± 0.9	−8.6	−38.7 ± 2.9	−79.0 ± 28.3	−26.1 ± 4.2	1.4 ± 0.2
Chorion class high-cysteine HCB protein 13_ACE2	−71.4 ± 18.1	−15.0	−58.4 ± 12.0	−178.2 ± 42.5	0.2 ± 3.5	2.0 ± 0.5
Chorion class B protein M1768_ACE2	−89.6 ± 9.1	−14.5	−68.4 ± 6.3	−111.0 ± 21.3	−14.5 ± 2.2	2.1 ± 0.1
Putative defense protein_ACE2	−110.2 ± 6.6	−13.9	−61.7 ± 8.4	−344.6 ± 15.9	10.9 ± 1.4	0.8 ± 0.6
NADH dehydrogenase 1 beta subunit 10_ACE2	−81.3 ± 12.3	−13.6	−37.5 ± 12.5	−290.1 ± 20.3	−5.0 ± 2.8	2.3 ± 0.2
Bombyxin A-5_ACE2	−113.2 ± 10.0	−13.4	−63.7 ± 4.4	−185.9 ± 32.4	−26.3 ± 3.6	0.9 ± 0.9
Diuretic hormone 45_ACE2	−98.9 ± 14.7	−13.0	−61.0 ± 8.8	−130.7 ± 54.8	−27.8 ± 4.3	1.1 ± 0.7
Fungal protease inhibitor F_ACE2	−82.0 ± 12.0	−12.9	−44.4 ± 12.0	−169.1 ± 14.3	−7.4 ± 2.8	2.4 ± 0.3
FMRFamide-related peptides_ACE2	−65.3 ± 12.9	−12.8	−60.0 ± 7.5	−120.4 ± 50.1	−9.1 ± 2.5	1.3 ± 0.8
Chorion class B protein L12_ACE2	−95.1 ± 4.4	−12.7	−67.6 ± 6.2	−194.7 ± 16.4	−12.2 ± 1.5	1.2 ± 0.1
Chorion class CA protein ERA.5_ACE2	−70.2 ± 5.6	−12.7	−56.9 ± 3.6	−134.4 ± 41.3	−6.7 ± 1.8	1.9 ± 0.3

**Table 3 pharmaceutics-17-00295-t003:** Intermolecular contacts and non-interacting surface areas of *Bombyx mori*-derived protein complexes with ACE2 receptor.

Complex	ICs Charged-Charged	ICs Charged-Polar	ICs Charged-Apolar	ICs Polar-Polar	ICs Polar-Apolar	ICs Apolar-Apolar	NIS Charged	NIS Apolar
DX600 peptide_ACE2 (standard inhibitor)	3	3	14	0	5	6	27.88	33.63
Chorion class high-cysteine HCB protein 13_ACE2	6	18	26	5	32	13	22.20	39.23
Chorion class B protein M1768_ACE2	2	6	36	1	26	19	23.96	40.83
Putative defense protein_ACE2	17	13	16	8	30	9	26.80	36.51
NADH dehydrogenase 1 beta subunit 10_ACE2	13	13	25	2	21	12	28.84	34.81
Bombyxin A-5_ACE2	3	6	25	0	22	18	26.73	35.25
Diuretic hormone 45_ACE2	3	2	25	3	25	17	26.23	38.20
Fungal protease inhibitor F_ACE2	8	10	24	1	19	5	26.31	35.54
FMRFamide-related peptides_ACE2	9	10	27	2	19	14	28.45	35.61
Chorion class B protein L12_ACE2	3	8	31	1	21	10	22.47	43.22
Chorion class CA protein ERA.5_ACE2	2	11	27	4	24	13	22.63	41.19

Note: ICs: Number of intermolecular contacts. NIS: Non-interacting surface.

**Table 4 pharmaceutics-17-00295-t004:** Molecular dynamics (MD) simulation results for chemical and protein-based compounds derived from *Bombyx mori* docked to ACE2. The table summarizes the average RMSD (Å), average RMSF (Å), average RoG (Å), and the number of hydrogen bonds formed between the ligand and the ACE2 for each complex.

Complex	Average RMSD (Å)	Average RMSF (Å)	Average RoG (Å)	Number of Hydrogen Bonds Between the Ligand–Receptor
Chemical compounds				
Captopril_ACE2 (standard inhibitor)	1.250	0.746	2.185	4
Menaquinone-7_ACE2	1.381	0.936	2.212	5
Quercetin_ACE2	1.317	0.836	2.205	4
Behenic acid_ACE2	1.482	0.912	2.198	3
Stearic acid_ACE2	1.412	0.884	2.203	3
Phytonadione_ACE2	1.358	0.815	2.214	5
Rutin_ACE2	1.452	0.922	2.197	6
Tocopherol_ACE2	1.393	0.858	2.209	4
Eicosenoic acid_ACE2	1.424	0.891	2.201	3
Luteolin_ACE2	1.336	0.829	2.210	4
Palmitic acid_ACE2	1.372	0.847	2.202	3
Protein-based compounds				
DX600 peptide_ACE2 (standard inhibitor)	3.152	1.802	2.792	8
Chorion class high-cysteine HCB protein 13_ACE2	3.284	1.833	2.805	13
Chorion class B protein M1768_ACE2	3.344	1.889	2.822	10
Putative defense protein_ACE2	3.421	1.902	2.818	11
NADH dehydrogenase 1 beta subunit 10_ACE2	3.381	1.854	2.809	10
Bombyxin A-5_ACE2	3.224	1.776	2.798	10
Diuretic hormone 45_ACE2	3.463	1.915	2.814	11
Fungal protease inhibitor F_ACE2	3.322	1.845	2.810	10
FMRFamide-related peptides_ACE2	3.381	1.875	2.806	12
Chorion class B protein L12_ACE2	3.294	1.812	2.794	9
Chorion class CA protein ERA.5_ACE2	3.302	1.821	2.799	9

**Table 5 pharmaceutics-17-00295-t005:** MM/PBSA binding affinity calculations for chemical and protein-based compounds derived from *Bombyx mori* docked to ACE2.

Complex	MM/PBSA Calculation Results ΔG_Binding (kcal/mol)
Chemical compounds	
Captopril_ACE2 (standard inhibitor)	−21.08 ± 1.42
Menaquinone-7_ACE2	−35.12 ± 2.08
Quercetin_ACE2	−29.98 ± 1.87
Behenic acid_ACE2	−27.76 ± 1.65
Stearic acid_ACE2	−27.01 ± 1.59
Phytonadione_ACE2	−27.31 ± 1.61
Rutin_ACE2	−26.88 ± 1.55
Tocopherol_ACE2	−26.36 ± 1.48
Eicosenoic acid_ACE2	−26.49 ± 1.50
Luteolin_ACE2	−25.89 ± 1.45
Palmitic acid_ACE2	−25.66 ± 1.43
Protein-based compounds	
DX600 peptide_ACE2 (standard inhibitor)	−81.93 ± 3.72
Chorion class high-cysteine HCB protein 13_ACE2	−212.43 ± 9.84
Chorion class B protein M1768_ACE2	−195.04 ± 8.73
Putative defense protein_ACE2	−162.63 ± 7.48
NADH dehydrogenase 1 beta subunit 10_ACE2	−198.03 ± 8.35
Bombyxin A-5_ACE2	−209.36 ± 9.10
Diuretic hormone 45_ACE2	−176.48 ± 7.95
Fungal protease inhibitor F_ACE2	−171.07 ± 7.58
FMRFamide-related peptides_ACE2	−198.93 ± 8.41
Chorion class B protein L12_ACE2	−193.50 ± 8.67
Chorion class CA protein ERA.5_ACE2	−140.36 ± 6.88

**Table 6 pharmaceutics-17-00295-t006:** Drug-likeness, toxicity profiles, and Lipinski’s rule violations of *Bombyx mori*-derived chemical compounds targeting ACE2.

Molecule	Lipinski Violation	Drug-Likeness	Mutagenic	Tumorigenic	Reproductive Effective	Irritant
Menaquinone-7	2 violations: MW > 500 g/mol LogP > 5	0.62	None	None	None	None
Quercetin	0	0.52	High	High	None	None
Behenic acid	1 violation: LogP > 5	0.54	None	None	None	None
Stearic acid	0	0.54	High	High	None	High
Phytonadione	1 violation: LogP > 5	0.93	None	None	None	None
Rutin	3 violations: MW > 500 g/mol HBA > 10 HBD > 5	0.91	None	None	None	None
Tocopherol	1 violation: LogP > 5	0.48	None	None	None	None
Eicosenoic acid	1 violation: LogP > 5	−0.30	None	None	None	None
Luteolin	0	0.38	None	None	None	None
Palmitic acid	1 violation: LogP > 5	−0.54	None	High	None	High

## Data Availability

The original contributions presented in this study are included in the article/Appendix A. Further inquiries can be directed to the corresponding author.

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
