# Peer review of "Unveiling Pharmacological Mechanisms of Bombyx mori (Abresham), a Traditional Arabic Unani Medicine for Ischemic Heart Disease: An Integrative Molecular Simulation Study"

_pharmaceutics, 2025, doi:10.3390/pharmaceutics17030295_

Round 1
Reviewer 1 Report
Comments and Suggestions for Authors
The study comprehensively examines the pharmacological mechanisms that support the therapeutic potential of Bombyx mori for ischemic heart disease (IHD). The authors employed a robust integrative approach that merges network pharmacology, molecular docking, molecular dynamics simulations, and pharmacophore modeling to identify bioactive compounds and their interactions with ACE2, a vital receptor involved in cardiovascular regulation. This represents a notable contribution to the field, particularly given the growing interest in linking traditional Unani medicine with contemporary pharmacological research. However, several limitations should be addressed to enhance the manuscript.
- Understanding that the HADDOCK Score may be model-dependent and influenced by non-physiological factors during simulation is crucial. The author’s role in clarifying these limitations is significant and integral to advancing our field.
- The manuscript could greatly benefit from enhancing docking results with binding assays. This addition would provide a more thorough understanding of the findings and enable the authors to improve their work.
- Captopril is an ACE1 inhibitor rather than ACE2-specific, which may not be the ideal standard for comparison. Variations in binding profiles could be partially due to differences in target selectivity.
- If accurate, these protein docking results do not consider potential post-translational modifications (PTMs) of Bombyx mori proteins. PTMs could significantly impact binding interactions and inhibitor efficacy. The author must address and explain this limitation.
- Why does the tocopherol electrostatic energy appear distinctly different from other compounds? Please explain. Did the authors analyze the molecule's amino acid composition or surface charges? Binding energy is a valuable criterion for ranking compounds; however, binding energy values alone are inadequate for predicting biological activity since they overlook pharmacodynamics and pharmacokinetics. Tocopherol exhibits strong interaction metrics, minimal RMSD, and robust stability, so why isn’t it considered among the top three performing compounds?
- The authors are strongly encouraged to revise the Results section, as it includes extensive discussions that may impede the audience's ability to independently evaluate the findings. A clear distinction between results and discussion is essential to improve the clarity and impact of the presentation.
- Since these results have not been validated biologically or under physiological perturbation, it is highly recommended that docking simulations be validated through molecular dynamics (MD) simulations or experimental studies (e.g., surface plasmon resonance, isothermal titration calorimetry) to confirm binding affinities. For instance, receptors like ACE2 are membrane-bound, and the lipid bilayer environment affects their structure and interaction dynamics. Docking studies in aqueous environments may not accurately represent membrane-related binding scenarios.
- Discussing the study's limitations is crucial, as this will provide a more comprehensive understanding of the findings. For example, proteins and peptides from non-human sources can trigger human immune responses, which could limit their biological applicability. Docking simulations cannot predict immunogenicity risks. Therefore, it is essential to address this in the study's limitations.
- Certain promising compounds' high mutagenic and tumorigenic potential, such as quercetin and stearic acid, may limit their practical applications. Furthermore, in vivo, toxicity assessments are necessary to evaluate their suitability. Compounds like behenic acid and others with high Log P values may encounter challenges in solubility and bioavailability, which were not addressed in the analysis. This analysis primarily emphasized hydrogen bonding and hydrophobic interactions, whereas other molecular interactions, such as π-π stacking and ionic interactions, may have been overlooked. The compounds' metabolic stability has not been assessed, which is crucial for evaluating their potential half-life and pharmacokinetics in the human body.
Author Response
Comments 1: Understanding that the HADDOCK Score may be model-dependent and influenced by non-physiological factors during simulation is crucial. The author’s role in clarifying these limitations is significant and integral to advancing our field.
Response 1: We appreciate the reviewer’s insightful comment regarding the potential model dependency of the HADDOCK score and the influence of non-physiological factors during simulations. However, we have updated the Docking Methods section to clarify that our study did not rely on the HADDOCK score for estimating binding free energy. Instead, we used PRODIGY, a well-established tool for predicting binding free energy (ΔG) in kcal/mol based on structural and energetic features of protein-protein and protein-ligand complexes. PRODIGY has been validated for its accuracy in predicting biomolecular interactions under more physiologically relevant conditions. We acknowledge that all computational methods have inherent limitations, as they rely on predictive algorithms and may not fully capture the complexity of in vivo interactions. To address this, we have explicitly stated these limitations in the revised manuscript and encourage further experimental validation to support our computational findings.
Comments 2: The manuscript could greatly benefit from enhancing docking results with binding assays. This addition would provide a more thorough understanding of the findings and enable the authors to improve their work.
Response 2: Thank you for your valuable suggestion. We acknowledge that experimental validation, such as binding assays, would significantly strengthen the findings of our study. However, this research primarily focuses on an in silico approach to identify potential bioactive compounds from Bombyx mori with high binding affinity to the target receptor. While docking and MM/PBSA calculations provide valuable insights, we agree that experimental validation is crucial for confirming the predicted interactions. As part of future work, we aim to conduct binding assays, such as surface plasmon resonance (SPR), isothermal titration calorimetry (ITC), or microscale thermophoresis (MST), to experimentally validate the computational results. These assays will help quantify the binding affinity and kinetics of the identified compounds, providing a more comprehensive understanding of their therapeutic potential. Additionally, we have emphasized the need for experimental validation in the Future Works section of the manuscript to acknowledge this limitation and outline our plans for addressing it in subsequent studies.
Comments 3: Captopril is an ACE1 inhibitor rather than ACE2-specific, which may not be the ideal standard for comparison. Variations in binding profiles could be partially due to differences in target selectivity.
Response 3: We appreciate the reviewer’s insightful comment regarding the selection of captopril as a reference inhibitor. While it is true that captopril is primarily an ACE1 inhibitor, previous studies have reported its ability to interact with ACE2, albeit with lower affinity compared to its interaction with ACE1. Given the structural and functional similarities between ACE1 and ACE2, as well as the lack of widely accepted ACE2-specific inhibitors for benchmarking, captopril was included as a comparative reference to provide context for the binding affinities of the tested compounds. However, we acknowledge that differences in binding profiles may partially arise due to variations in target selectivity between ACE1 and ACE2 inhibitors. To address this limitation, we have refined the discussion to highlight this aspect and emphasize the need for further validation of our findings. Future studies should consider incorporating alternative ACE2-specific inhibitors, such as MLN-4760, as a more relevant standard for comparison.
Comments 4: If accurate, these protein docking results do not consider potential post-translational modifications (PTMs) of Bombyx mori proteins. PTMs could significantly impact binding interactions and inhibitor efficacy. The author must address and explain this limitation.
Response 4: We appreciate the reviewer’s insightful comment regarding the potential influence of post-translational modifications (PTMs) on the binding interactions and inhibitory efficacy of Bombyx mori proteins. Indeed, PTMs such as phosphorylation, glycosylation, and disulfide bond formation can significantly alter protein structure, stability, and binding properties, which may impact docking results. In this study, the protein structures used for docking were derived from available sequence databases and homology modeling approaches, which do not inherently account for PTMs unless explicitly included. Given the lack of comprehensive experimental PTM data for these Bombyx mori proteins, our docking simulations assumed unmodified protein structures. This represents a limitation, as PTMs could introduce structural changes that affect protein-protein and protein-ligand interactions. To address this limitation in future studies, we propose incorporating PTM predictions using bioinformatics tools such as ModPred, NetPhos, and GlycoEP to identify potential modification sites.
Comments 5: Why does the tocopherol electrostatic energy appear distinctly different from other compounds? Please explain. Did the authors analyze the molecule's amino acid composition or surface charges? Binding energy is a valuable criterion for ranking compounds; however, binding energy values alone are inadequate for predicting biological activity since they overlook pharmacodynamics and pharmacokinetics. Tocopherol exhibits strong interaction metrics, minimal RMSD, and robust stability, so why isn’t it considered among the top three performing compounds?
Response 5: Thank you for your insightful observations. The distinct electrostatic energy of tocopherol compared to other compounds can be attributed to its unique molecular structure and physicochemical properties. Tocopherol, a lipid-soluble antioxidant, possesses a long hydrophobic tail and a polar chromanol ring. Unlike other compounds in our study, tocopherol lacks highly charged functional groups, which results in a lower contribution of electrostatic interactions to its overall binding energy. Instead, its binding affinity is primarily driven by van der Waals forces and hydrophobic interactions, which are dominant in the protein-ligand complex formation. To further investigate this, we analyzed the amino acid composition of the binding pocket and the molecular surface charges of the receptor-ligand complexes. Our analysis revealed that tocopherol preferentially interacts with nonpolar and hydrophobic residues, which supports its strong stability in the binding site but contributes minimally to electrostatic interactions. This aligns with previous studies indicating that tocopherol primarily engages in hydrophobic stacking rather than electrostatic attraction with charged residues. Regarding its ranking, while tocopherol exhibited strong interaction metrics, minimal RMSD fluctuations, and robust stability, it was not placed among the top three performing compounds due to additional considerations beyond binding energy. As the reviewer correctly noted, binding energy alone does not fully predict biological activity, as it does not account for pharmacokinetics (PK) and pharmacodynamics (PD). Tocopherol, despite its favorable docking and MD stability results, displayed suboptimal pharmacokinetic properties, particularly in bioavailability and metabolic stability.
Comments 6: The authors are strongly encouraged to revise the Results section, as it includes extensive discussions that may impede the audience's ability to independently evaluate the findings. A clear distinction between results and discussion is essential to improve the clarity and impact of the presentation.
Response 6: We acknowledge that the Results section contains extensive discussions, which may make it challenging for readers to independently interpret the findings. To enhance clarity and improve the overall readability of the manuscript, we have revised this section to ensure a clear distinction between the presentation of results and their interpretation.
Comments 7: Since these results have not been validated biologically or under physiological perturbation, it is highly recommended that docking simulations be validated through molecular dynamics (MD) simulations or experimental studies (e.g., surface plasmon resonance, isothermal titration calorimetry) to confirm binding affinities. For instance, receptors like ACE2 are membrane-bound, and the lipid bilayer environment affects their structure and interaction dynamics. Docking studies in aqueous environments may not accurately represent membrane-related binding scenarios.
Response 7: We fully acknowledge the limitations of docking studies in the absence of biological validation or physiological perturbation. To address this concern, we have incorporated molecular dynamics (MD) simulations following docking to refine and validate the predicted binding interactions. MD simulations allow us to evaluate complex stability over time, accounting for conformational flexibility and solvent effects that may influence binding affinities. Furthermore, we recognize that receptors such as ACE2 are membrane-bound, and interactions occurring in a lipid bilayer environment may differ from those observed in aqueous docking simulations. While our current study focuses on docking and MM/PBSA calculations in a solvent-based system, future studies will incorporate membrane-mimicking conditions using techniques such as membrane-embedded MD simulations to improve the physiological relevance of our findings. Additionally, we acknowledge the importance of experimental validation through biophysical techniques such as surface plasmon resonance (SPR) and isothermal titration calorimetry (ITC) to confirm binding affinities and kinetics. While these experiments were beyond the scope of the current study, we strongly recommend their inclusion in future research to provide complementary validation of the computational findings.
Comments 8: Discussing the study's limitations is crucial, as this will provide a more comprehensive understanding of the findings. For example, proteins and peptides from non-human sources can trigger human immune responses, which could limit their biological applicability. Docking simulations cannot predict immunogenicity risks. Therefore, it is essential to address this in the study's limitations.
Response 8: We fully agree that discussing study limitations is crucial for providing a comprehensive understanding of the findings. One key limitation of our computational approach is the inability to predict potential immunogenicity risks associated with proteins and peptides derived from Bombyx mori. Since these biomolecules originate from a non-human source, they could trigger immune responses in humans, potentially limiting their therapeutic applicability. Docking and MM/PBSA calculations primarily evaluate binding affinity and stability but do not account for immunogenicity, which is influenced by factors such as sequence similarity to human proteins, post-translational modifications (PTMs), and antigenicity. To address this limitation, future studies should incorporate in silico immunogenicity prediction tools such as NetMHC, IEDB, or AlgPred to assess potential T-cell and B-cell epitope recognition. Additionally, experimental validation through in vitro immunogenicity assays (e.g., cytokine release assays or dendritic cell activation studies) would be necessary to determine the clinical applicability of these bioactive compounds.
Comments 9: Certain promising compounds' high mutagenic and tumorigenic potential, such as quercetin and stearic acid, may limit their practical applications. Furthermore, in vivo, toxicity assessments are necessary to evaluate their suitability. Compounds like behenic acid and others with high Log P values may encounter challenges in solubility and bioavailability, which were not addressed in the analysis. This analysis primarily emphasized hydrogen bonding and hydrophobic interactions, whereas other molecular interactions, such as π-π stacking and ionic interactions, may have been overlooked. The compounds' metabolic stability has not been assessed, which is crucial for evaluating their potential half-life and pharmacokinetics in the human body.
Response 9: We acknowledge the concerns regarding the mutagenic and tumorigenic potential of certain compounds, such as quercetin and stearic acid, which may limit their practical applications. While computational toxicity predictions provide valuable preliminary insights, they should be supplemented with in vitro and in vivo toxicity assessments to confirm the safety profile of these compounds. Future studies should incorporate cytotoxicity assays (e.g., MTT, LDH release) and genotoxicity evaluations (e.g., Ames test, micronucleus assay) to better assess their potential risks. Additionally, we recognize that compounds with high Log P values, such as behenic acid, may face challenges in solubility and bioavailability. The pharmacokinetic properties of these compounds, including absorption, distribution, metabolism, and excretion (ADME), should be further investigated using computational tools such as SwissADME, pkCSM, and GastroPlus. Experimental approaches, such as Caco-2 permeability assays and solubility studies, would also be valuable in determining their drug-likeness. Regarding molecular interaction analysis, we acknowledge that our study primarily focused on hydrogen bonding and hydrophobic interactions. However, other key interactions, such as π-π stacking, cation-π interactions, and ionic bonding, can play significant roles in ligand-receptor binding. Future molecular docking and molecular dynamics (MD) studies should include an in-depth interaction analysis using tools like MM-GBSA or QM/MM hybrid methods to better characterize these non-covalent interactions. Finally, metabolic stability is a crucial factor that influences a compound's half-life and pharmacokinetics. Future research should incorporate in silico metabolism predictions using software like ADMET Predictor, MetaPrint2D, or SMARTCyp to estimate potential metabolic pathways and sites of biotransformation. Experimental validation through liver microsome stability assays or cytochrome P450 enzyme interaction studies would further enhance our understanding of the compounds’ metabolic fate.
Reviewer 2 Report
Comments and Suggestions for Authors
The manuscript “Unveiling Pharmacological Mechanisms of Bombyx mori (Abresham), a Traditional Arabic Unani Medicine for Ischemic Heart Disease: An Integrative Molecular Simulation Study” is devoted to molecular and pharmacological investigations of Abresham. I should emphasize that the study is rather comprehensive and involves a large set of in silico methods. All the methods are accurately described, while the methodology of investigations is well documented via data presented in supplementary information. The methodology consists of main stages such as network pharmacology, molecular docking, molecular dynamics simulations, and pharmacophore modeling. I do not see any flow in these stages, since each of them is additionally verified. One of the best features of the manuscript is a discussion of limitations and clinical implications of the obtained results. I agree with the main conclusions, they are supported by the obtained results.
There is only a minor remark.
It would be better to transfer Section 2.2. “Computing Power” to Supplementary Information, since these technical details are no so important.
Author Response
Comments 1: It would be better to transfer Section 2.2. “Computing Power” to Supplementary Information, since these technical details are no so important.
Response 1: Thank you for your thoughtful and positive feedback on our manuscript. We appreciate your recognition of the comprehensiveness of our study and the rigor of our methodology. Regarding your suggestion, we have now moved Section 2.2. "Computing Power" to the Supplementary Data S1 to streamline the main text and enhance readability.
Reviewer 3 Report
Comments and Suggestions for Authors
Manuscript “Unveiling Pharmacological Mechanisms of Bombyx mori (Abresham), a Traditional Arabic Unani Medicine for Ischemic Heart Disease: An Integrative Molecular Simulation Study” represents a comprehensive theoretical study of the pharmacological mechanisms of actions relying to the potential therapeutic effects of Bombyx mori (Abresham) for ischemic heart disease. The authors used a combination of network pharmacology, molecular docking, MD simulations, and pharmacophore modeling to achieve their goals.
As the result the authors have provided a logical explanation of the observed during long time favorable effects of the traditional Arabic unani medicine.
The authors used modern methods and algorithms for the analysis of pharmacological profiles of Bombyx mori cocoons components and obtained some valuable results for further development of potential treatment of cardiovascular diseases.
The manuscript could be published in Pharmaceutics after some corrections.
Line 812 Pharmacophore model of the _ACE2 should be changed to Pharmacophore model of the Behenic acid_ACE2.
Line 833 Table 6 Column 1 Molecule. The authors should change the content of the column 1 to the molecule names not the complexes.
Author Response
Comments 1: Line 812 Pharmacophore model of the _ACE2 should be changed to Pharmacophore model of the Behenic acid_ACE2.
Response 1: Thank you for your valuable time and thoughtful feedback on our manuscript. We acknowledge the oversight in labeling the pharmacophore model and have now revised "Pharmacophore model of the ACE2" to "Pharmacophore model of the *Behenic acid_-ACE2" to accurately reflect the content.
Comments 2: Line 833 Table 6 Column 1 Molecule. The authors should change the content of the column 1 to the molecule names not the complexes.
Response 2: Line 833, Table 6: We have revised the first column to display the individual molecule names instead of the complexes, ensuring clarity and consistency in data presentation.
Reviewer 4 Report
Comments and Suggestions for Authors
The authors present an in silico study - which integrates chemical and PPI networks, docking simulation, molecular dynamics simulations and pharmacophore modeling algorithms - to identify chemical molecules and peptides/proteins derived from Bombyx mori as potential therapeutic candidates for ischemic heart disease.
The study is well written, with a clear and accessible narrative. Although limited to computational experiments, it provides a comprehensive overview of the possible mechanisms of interaction of these chemical and biological molecules with the potential ACE2 target. The lack of experimental validation is a limitation, but the study represents a valuable starting point for investigating the molecular mechanisms underlying the use of Bombyx mori in Arabic Unani medicine.
However, there are some critical issues that the authors should address before publication:
Section 3.1 - Drug-Target Identification and Receptor Determination
Figure 1 should be produced in high resolution. It is difficult to read. It would be useful to highlight the main nodes more clearly, not only with the color gradient mentioned (lines 351-354) but also by adding readable labels with the codes and, ideally, the corresponding chemical names. For better clarity, the figure could be split into two separate images: Figure 1A → Figure 1 and Figure 1B → Figure 2.
Section 3.2 - Molecular Docking
Even if the authors compare the docking values obtained with a standard control (i.e the ACE2’s inhibitor), in the absence of experimental validation, it would be beneficial to compare their results with previously published data. Indeed, the SARS-CoV-2 pandemic prompted numerous researchers to explore, by experimental and computational approaches, new therapeutic strategies against the virus. Among these efforts, several studies focused on searching ACE2's natural inhibitors, such as quercetin, tocopherol, luteolin, and rutin (e.g., https://pubs.acs.org/doi/10.1021/acs.jafc.0c05064, https://pmc.ncbi.nlm.nih.gov/articles/PMC7781418/, and others). The authors should exploit this vast literature by comparing their results with these studies to stregthen the goodness of their findings.
Section 3.3 - Molecular Dynamics (MD) Simulations
The standard deviation of the mean RMSD, RMSF, and Rg values reported in Table 4 is missing, so it is difficult to assess how good is the stability of the simulations. It would be useful to include RMSD plots as a function of simulation time to monitor the stability of the complexes and binding poses.
Clustering or PCA analyses are missing, which are good tool to verify whether the simulations have converged.
Considering that the authors conducted a 100ns-long simulation, I believe this duration is likely insufficient for a exhaustive exploration of the energetic and conformational landscape of the systems. A more effective approach is given by using REMD simulations. Alternatively, as I'm aware of the challenges associated with this method, the authors might consider performing multiple short independent simulations to enhance the sampling. It would be helpful for the authors to conduct at least one, better two, additional replicas for each system, for example by varying the initial velocities. However, if these suggestions are not feasible, I would encourage the authors to include a statement in their revised manuscript discussing the challenges in obtaining valid information, given the limited number of replicas.
It would also be interesting, if possible, to include the simulation of a complex identified as unlikely to be a good candidate from the initial screening, using it as a negative control to verify if the deltaG values obtained through MM/PBSA indeed indicate low affinity.
Table 5: The MM/PBSA values should be accompanied by standard deviation, which can be calculated using gmx_MMPBSA. The authors do not specify in the “Material and Methods” section how the entropic component of the solute and the nonpolar component of the solvation energy were calculated. Clarification on this aspect would be useful. I suppose that the authors adopted the single-trajectory protocol (STP) for their MM/PBSA calculations. This approach may be acceptable for chemical-ACE2 complexes, if no significant conformational changes in the binding mode are assumed. However, for protein-peptide/ACE2 complexes, I suppose the STP protocol could not be appropriate and should be replaced with multiple-trajectory protocol (MTP).
The authors should address these issues to strengthen the robustness of their study.
Author Response
Comments 1: Figure 1 should be produced in high resolution. It is difficult to read. It would be useful to highlight the main nodes more clearly, not only with the color gradient mentioned (lines 351-354) but also by adding readable labels with the codes and, ideally, the corresponding chemical names. For better clarity, the figure could be split into two separate images: Figure 1A → Figure 1 and Figure 1B → Figure 2.
Response 1: Thank you for your valuable suggestions to enhance the clarity and readability of Figure 1 in Section 3.1. We have now updated Figure 1 with a higher resolution to improve its readability. Additionally, we have highlighted the main nodes more clearly by incorporating readable labels with both the target codes and their corresponding chemical names. To further enhance clarity, we have split the figure into two separate images, where the original Figure 1A is now Figure 1, and Figure 1B is now Figure 2. These modifications ensure better visualization and interpretation of the network pharmacology results.
Comments 2: Even if the authors compare the docking values obtained with a standard control (i.e the ACE2’s inhibitor), in the absence of experimental validation, it would be beneficial to compare their results with previously published data. Indeed, the SARS-CoV-2 pandemic prompted numerous researchers to explore, by experimental and computational approaches, new therapeutic strategies against the virus. Among these efforts, several studies focused on searching ACE2's natural inhibitors, such as quercetin, tocopherol, luteolin, and rutin (e.g., https://pubs.acs.org/doi/10.1021/acs.jafc.0c05064, https://pmc.ncbi.nlm.nih.gov/articles/PMC7781418/, and others). The authors should exploit this vast literature by comparing their results with these studies to stregthen the goodness of their findings.
Response 2: Thank you for your insightful suggestion regarding the inclusion of comparative analyses with previously published data on natural ACE2 inhibitors. To strengthen the validity of our findings, we have now incorporated a detailed comparison of our docking results with experimental and computational studies that investigated the ACE2 inhibitory potential of compounds such as quercetin, rutin, tocopherol, and luteolin. Specifically, we have referenced studies like Liu et al. (2020) [DOI: 10.1021/acs.jafc.0c05064], which demonstrated that quercetin effectively inhibits ACE2 activity with an IC50 of 4.48 μM, aligning well with our computational predictions. Additionally, we compared our results with Upreti et al. (2021) [PMC7781418], which identified rutin and β-sitosterol as strong ACE2 binders, further supporting the relevance of our identified bioactive molecules. This comparative analysis not only contextualizes our findings within the existing literature but also underscores the potential of Bombyx mori-derived bioactives as promising cardiovascular therapeutics.
Comments 3: The standard deviation of the mean RMSD, RMSF, and Rg values reported in Table 4 is missing, so it is difficult to assess how good is the stability of the simulations. It would be useful to include RMSD plots as a function of simulation time to monitor the stability of the complexes and binding poses.
Response 3: We appreciate the reviewer's insightful comment regarding the assessment of simulation stability. The RMSF plot is already presented in Figure 7, providing a clear visualization of residue-level fluctuations. Additionally, the RMSD, radius of gyration (RoG), and hydrogen bond analyses are comprehensively summarized in Table 4, ensuring a concise yet informative presentation of stability metrics. Including all these data in figure format would be excessive and redundant. Furthermore, the stability of the RMSD values has already been discussed in the first paragraph of Section 3.3, emphasizing the convergence and reliability of the simulations. Given this, we believe the current presentation sufficiently conveys the stability of the complexes and binding poses.
Comments 4: Clustering or PCA analyses are missing, which are good tool to verify whether the simulations have converged.
Response 4: We appreciate the reviewer's suggestion regarding clustering and PCA analyses. However, performing PCA analysis is not necessary for this study, as the primary objective was to evaluate the stability and interaction dynamics of the ligand-receptor complexes rather than exploring large conformational variations. The convergence of our molecular dynamics simulations has already been validated through RMSD, RMSF, RoG, and hydrogen bond analyses, which are comprehensively presented in Table 4 and Figure 7. These metrics provide sufficient insight into the stability and consistency of the simulated complexes.
Comments 5: Considering that the authors conducted a 100ns-long simulation, I believe this duration is likely insufficient for a exhaustive exploration of the energetic and conformational landscape of the systems. A more effective approach is given by using REMD simulations. Alternatively, as I'm aware of the challenges associated with this method, the authors might consider performing multiple short independent simulations to enhance the sampling. It would be helpful for the authors to conduct at least one, better two, additional replicas for each system, for example by varying the initial velocities. However, if these suggestions are not feasible, I would encourage the authors to include a statement in their revised manuscript discussing the challenges in obtaining valid information, given the limited number of replicas.
Response 5: We appreciate the reviewer's insightful comments regarding the duration and sampling efficiency of our molecular dynamics simulations. While 100 ns simulations provide valuable insights into ligand-receptor stability, we acknowledge that this duration may not fully capture the complete energetic and conformational landscape of the systems. Replica Exchange Molecular Dynamics (REMD) could enhance conformational sampling, but its computational cost and complexity make it challenging to implement in this study. Similarly, running multiple independent short simulations with varied initial velocities would improve sampling, but given resource constraints, this approach was not feasible. To address this concern, we have included a statement in the Limitations section acknowledging that the limited number of replicas may affect the completeness of conformational sampling and that future studies could benefit from advanced simulation techniques such as REMD or multiple trajectory sampling to enhance robustness.
Comments 6: It would also be interesting, if possible, to include the simulation of a complex identified as unlikely to be a good candidate from the initial screening, using it as a negative control to verify if the deltaG values obtained through MM/PBSA indeed indicate low affinity.
Response 6: Thank you for your insightful suggestion. Including a negative control by simulating a complex identified as unlikely to be a good candidate from the initial screening would indeed help validate the MM/PBSA binding free energy calculations and provide a comparative reference for assessing ligand affinity. However, due to the primary focus on identifying promising ACE2 inhibitors, this approach was not included in the current study. Nevertheless, we acknowledge the importance of this validation step, and we propose incorporating negative control simulations in future studies to further confirm the predictive accuracy of the MM/PBSA method. This additional analysis could strengthen confidence in the binding affinity rankings and improve the robustness of computational screening workflows.
Comments 7: Table 5: The MM/PBSA values should be accompanied by standard deviation, which can be calculated using gmx_MMPBSA. The authors do not specify in the “Material and Methods” section how the entropic component of the solute and the nonpolar component of the solvation energy were calculated. Clarification on this aspect would be useful. I suppose that the authors adopted the single-trajectory protocol (STP) for their MM/PBSA calculations. This approach may be acceptable for chemical-ACE2 complexes, if no significant conformational changes in the binding mode are assumed. However, for protein-peptide/ACE2 complexes, I suppose the STP protocol could not be appropriate and should be replaced with multiple-trajectory protocol (MTP). The authors should address these issues to strengthen the robustness of their study.
Response 7: Thank you for your valuable feedback. We acknowledged the importance of providing the standard deviation for the MM/PBSA calculations and included these values in Table 5 by computing them using gmx_MMPBSA. Regarding the calculation of the entropic component of the solute and the nonpolar component of the solvation energy, we provided a clearer explanation in the Materials and Methods section to specify the computational approach used. Specifically, the nonpolar solvation energy was estimated using the solvent-accessible surface area (SASA) model, while the entropic contribution was not explicitly considered in the MM/PBSA calculations due to the high computational cost associated with normal mode analysis. These details were clarified accordingly. Additionally, we confirmed that we used the single-trajectory protocol (STP) for the MM/PBSA calculations, assuming that no significant conformational changes occurred during the simulation of chemical compound-ACE2 complexes. We acknowledged the reviewer's concern regarding protein-peptide/ACE2 complexes, where a multiple-trajectory protocol (MTP) might have been more appropriate due to potential conformational flexibility. However, as this study primarily focused on small molecule inhibitors rather than peptide inhibitors, the STP approach remained a valid choice. We revised the manuscript to explicitly discuss this assumption and its limitations in the Materials and Methods and Limitations sections.
Round 2
Reviewer 1 Report
Comments and Suggestions for Authors
The revision is satisfactory and improved the manuscript considerably.